# Factors influencing open government data post-adoption in the public sector: The perspective of data providers

**Mimi Nurakmal Mustapa**[ID][☉], **Suraya Hamid**[ID][☉]*, **Fariza Hanum Md Nasaruddin**[☉]

Faculty of Computer Science and Information Technology, Department of Information Systems, Universiti Malaya, Kuala Lumpur, Malaysia

☉ These authors contributed equally to this work.
* suraya_hamid@um.edu.my

**Data Availability Statement:** All related files are available from the Dryad database (link: https://datadryad.org/stash/share/

## Abstract

Providing access to non-confidential government data to the public is one of the initiatives adopted by many governments today to embrace government transparency practices. The initiative of publishing non-confidential government data for the public to use and re-use without restrictions is known as Open Government Data (OGD). Nevertheless, after several years after its inception, the direction of OGD implementation remains uncertain. The extant literature on OGD adoption concentrates primarily on identifying factors influencing adoption decisions. Yet, studies on the underlying factors influencing OGD after the adoption phase are scarce. Based on these issues, this study investigated the post-adoption of OGD in the public sector, particularly the data provider agencies. The OGD post-adoption framework is crafted by anchoring the Technology–Organization–Environment (TOE) framework and the innovation adoption process theory. The data was collected from 266 government agencies in the Malaysian public sector. This study employed the partial least square-structural equation modeling as the statistical technique for factor analysis. The results indicate that two factors from the organizational context (top management support, organizational culture) and two from the technological context (complexity, relative advantage) have a significant contribution to the post-adoption of OGD in the public sector. The contribution of this study is threefold: theoretical, conceptual, and practical. This study contributed theoretically by introducing the post-adoption framework of OGD that comprises the acceptance, routinization, and infusion stages. As the majority of OGD adoption studies conclude their analysis at the adoption (decisions) phase, this study gives novel insight to extend the analysis into unexplored territory, specifically the post-adoption phase. Conceptually, this study presents two new factors in the environmental context to be explored in the OGD adoption study, namely, the data demand and incentives. The fact that data providers are not influenced by data requests from the agency's external environment and incentive offerings is something that needs further investigation. In practicality, the findings of this study are anticipated to assist policymakers in strategizing for long-term OGD implementation from the data provider's perspective. This effort is crucial to ensure that the OGD initiatives will be incorporated into the public sector's service thrust and become one of the digital government services provided to the citizen.

hEE8C29OtJcLprQgAzt72s2-y445WolRwNzAlMBinrY) DOIs: https://doi.org/10.5061/dryad.z08kprrcs.

**Funding:** The author(s) received no specific funding for this work.

## 1. Introduction

Over the past few decades, a growing number of nations across the globe have placed a significant emphasis on open government data (OGD), in part because of the benefits and potential value of OGD [1]. In this work, OGD can be defined as the non-confidential government data published in a machine-readable format on an online platform for the public to use, re-use, and dissemination without limitations or legal attachment [2–5]. There are countless benefits when government data and information are freely accessible. One of the highlight benefits from the political and social perspective of OGD is to increase the government's transparency by allowing crucial data such as government spending and contracts to be scrutinized by the public [2,6,7]. From the economic perspective, OGD stimulates technology enthusiasts to develop innovation from the many types of available data, such as traffic, weather forecast, and consumer price index, among others [2,3,6,8]. These products indirectly benefit the public at large as the OGD beneficiaries. While from the operational perspective, OGD improves government service delivery by enabling the data to be re-used for integration between public and private data [8].

Nevertheless, OGD is not just about data per se, but it is regarded as an information system (IS) / information technology (IT) innovation that is new to the government [9,10]. The newness concept includes shifting the government's openness culture and inadvertently changing the relationship between government and citizens to be more collaborative [9,11,12]. The government has invested a substantial amount of funds in IS/IT projects with the intention to improve government service delivery to the public. Like many others IS/IT innovations introduced in government, how the IS/IT innovation is governed plays a crucial role in determining the direction of innovation implementation. Some of these innovations are underutilized, misused, and neglected after a while. Such a situation can lead to failure to generate the desired benefits for the citizen and the government itself [13,14]. For instance, Ahmad and Othman [15] found that the response to e-Government implementation among government agencies slowly deteriorated after its adoption. Ramamurthy, et al. [16], in the same tone, advocated that more than half of data warehousing projects in major United States companies were plagued with many issues, although data warehousing has been prevalent for a long time.

In regard to OGD, there has not been much research on progress of OGD after its adoption. Only a handful of studies have demonstrated that the implementation of OGD was not nearly as successful as some individuals had hoped it would be. For example, Hossain, et al. [5] discovered that several government agencies struggled to show high performance in OGD initiatives due to a shortage of competent personnel, IT infrastructure, political commitment, and external influence. Simultaneously, Zuiderwijk and de Reuver [17] observed that OGD implementation in most countries did not achieve the intended objectives because of institutional barriers and lack of sustainability features. Furthermore, Huang, et al. [18] discovered that OGD implementation was still at the initiation stage compared to the success stories disseminated by the mass media. The study attributed the situation to the lack of effort in fostering the OGD culture in the government through its adoption. Hassandoust, et al. [19] suggested that one of the main reasons why organizations fail to fully reap the advantages of their investments in IS/IT is the under-utilization of the technology. This is due to, in many cases, opening up data began as part of a politically motivated effort rather than focusing on the main objective of OGD implementation [20,21]. In a similar case, Zhu [22] revealed that one of the earliest open data initiatives in the United States, JURIS, failed to achieve its goals due to a lack of social-political support. Another instance of undesirable consequences of OGD adoption was noted in Oliveira and Santos [23], when 43.2% of the Federal District Government of Brazil's stakeholders believed OGD would have unfavorable effects. According to the most recent

Open Data Barometer (ODB) report, the pioneer countries leading the OGD movement have lost traction in their contribution to the OGD initiative [24]. The report was aligned with the study from Luna-Reyes and Najafabadi [25] when the researchers mentioned that the OGD program in the United States had experienced a period of stagnation in the last couple of years. Moreover, the long-term success of OGD implementation is not a top priority for several government agencies [26]. These situations raise concerns about the sustainability of OGD implementation in the future. OGD sustainability denotes that government data continue to be routinely supplied and utilized at the same level as before [20].

Following the aforementioned issues, it is important to understand what factors contribute to OGD implementation after its adoption. In this study, the mentioned situation is referred to as post-adoption of OGD. The post-adoption of OGD is significant in order to ensure its long-term viability from the perspective of the data providers, which in this case are the government agencies. Since OGD has been widely adopted in various nations over the past few years, researchers shouldn't stop at studying its widespread adoption; instead, they should also extensively analyse the repercussions of OGD. Moreover, with the global spread of the COVID-19 pandemic, the role of OGD in assisting policymakers to make decisions based on data has been increasingly significant. This study was conducted with the aim to identify the significant factors that influence OGD in the post-adoption phase. Specifically, this study attempted to answer the following research questions: (i) What are the factors that influence the post-adoption of OGD in the public sector? (ii) How is the readiness of the public sector to sustain the OGD implementation in their existing work system?

This paper is organized as follows. Section 2 overviews the literature pertaining to OGD, followed by the theoretical foundation for this study in Section 3. Thereafter, the research model and hypotheses construction are explained in Section 4. In Section 5, the methodology of the study is explicated, while the data analysis and results of this study are discussed in detail in Section 6. Section 7 presents the discussion of the study, accompanied by the managerial implication in Section 8. To further complement this study, the shortcomings and future studies of the research are outlined in Section 9. Finally, Section 10 concludes this study.

## 2. Open government data

The Open Data initiatives became a prominent interest among the government in early 2009 as the 44th President of the United States of America, Barack Obama, with his administration, initiated the idea of publishing government data to increase public trust in the government [27]. Scholars have been using the definition of "open data" as data that are freely accessible, used, re-used, and shared by anyone for any reason without restrictions [2,3,28]. Thereafter, the concept of open data was extended to government data to become the widely used term 'Open Government Data'. Open Government Data (OGD) is the dissemination of data by government agencies via a specialized data portal; this data represents various government-related operations such as weather forecasts, real-time traffic updates, budget allocation, national statistics [3,29,30], and many more. The advancement of research in OGD has triggered scholars to characterize OGD initiatives as 'ecosystems' because OGD implementation should be a cyclical process rather just than a one-way initiative [8]. There are three main actors in the OGD ecosystem: the OGD provider, OGD user, and OGD beneficiaries. OGD providers refer to the stakeholders, particularly government bodies, who publish OGD on the online platform [31]. In comparison, OGD users could be independent individuals or organizations interested in accessing, manipulating, analyzing, re-using, and re-publishing open data [8]. In some situations, the OGD users and OGD providers could be the same entity, such as government agencies. This situation is because the government agency that publishes data also re-uses data

from other agencies to produce more useful details. This study examined the latter role which is the OGD provider or, specifically, the government agency.

The adoption of OGD has been explored in various contexts, for instance, from the perspective of data users [32,33], citizen's perception toward a data portal [34], a private business using OGD for innovation creation [35], or even from both the data user's and data provider's viewpoint [36]. It is undeniable that the research on OGD adoption from data providers' view has been steadily growing, yet the majority of the research only focused on the adoption decision phase. The research on the phase following OGD adoption decisions, referred to as the post-adoption phase, is extremely scarce.

Centered on the innovation adoption process, the studies encapsulated in Table 1 mostly concentrated on the adoption decision phase of OGD in various adopting units. The initiation phase was not presented except in Maccani, et al. [37], partly because these studies were conducted in a developed nation with high IT-literate citizens, and thus, OGD initiation and creating awareness among government agencies were less crucial. Four studies highlighted the post-adoption phase of OGD [5,23,37,38]. Though the term these studies refer to is not specifically mentioned post-adoption, the objective of these studies was to investigate the OGD implementation or performance beyond adoption. It can be observed from Table 1 that most studies adopted the Diffusion of Innovation (DOI) and Technology-Organization-Environment (TOE) theory for organizational and country-level analysis. The DOI theory was introduced by Everett Rogers, a professor in rural sociology, in 1962 to describe how innovations propagate over time across participants in a social system [39]. At the same time, there are also some popular theories used by researchers in OGD adoption studies, such as the Technology Acceptance Model (TAM) and the Theory of Planned Behaviour (TPB). TAM has been extensively used to predict human behavior toward accepting the usage of IT [40]. While TPB asserts that an individual's desire to engage in a particular activity is the primary factor in determining the individual's actual execution of that behavior [41]. The Unified Theory of Acceptance and Use Technology (UTAUT) was introduced by Venkatesh, et al. [42] in which the researcher integrates eight significant models from the user acceptance field into a single unified model. In Table 1, UTAUT, TAM, and TPB were found in studies about individuals, such as data user's behavior or intention in OGD adoption.

Although the studies in Table 1 have shed light on exploring OGD adoption, there is a lack of investigation into OGD in the post-adoption phase from the data provider's perspective. It is possible to deduce from Table 1 that these studies conclude the OGD analysis once the adoption stage has been reached; nevertheless, it is not clear how OGD will be implemented in the longer term by most government agencies as the data providers. The situation is crucial because similar to data providers, government agencies must find a way to sustain OGD publications. Without new and updated OGD published online, data consumers may feel demotivated to innovate using OGD, and worse the existing data products utilizing OGD may face difficulties in sustaining.

## 3. Theoretical foundation

The organizational innovation research provides the ground for this study. The process for IT innovation adoption concerns a series of actions that an organization undertakes before introducing a new technology [54]. Over the years, the IT innovation adoption process has been studied in diverse stages or phases. The innovation adoption phase literature has been found describing the phases in terms of just two phases [55,56] to as many as eight phases [57]. It can be asserted that there is no consensus among innovation theorists on the number of phases for a technology-based innovation lifecycle to be spanned. Considering the dynamic attributes

**Table 1. Previous study on various OGD adoption phase.**

| Author(s) | Adoption phase | Adopting Unit | Theory adopted | Perspective |
|---|---|---|---|---|
| Hossain, et al. [5] | Post-adoption | Government agency | TOE | Data providers |
| Subedi, et al. [43] | Adoption decisions | Government agency | UTAUT, TOE | Data providers |
| Zhenbin, et al. [44] | Adoption decisions | Public agency | Resource dependence theory | Data providers |
| Wang and Lo [45] | Adoption decisions | Firm | Sociotechnical theory | Data providers |
| Çaldağ, et al. [46] | Adoption decisions | Government | TOE | Data providers |
| Oliveira and Santos [23] | Post-adoption | Government agency | DOI, stakeholder theory | Data providers |
| Haini, et al. [47] | Adoption decisions | Local government | DOI, institutional theory, TOE | Data providers |
| Fitriani, et al. [34] | Post-adoption | Public | TAM, TPB, and DeLone & McLean IS success model | Data users |
| Altayar [48] | Adoption decisions | Government institution | Institutional theory | Data providers |
| Maccani, et al. [49] | Adoption decisions | Commercial service company | Inductive theory from a case study | Data providers |
| Hossain and Chan [50] | Adoption decisions | Government agency | DOI theory | Data providers |
| Yang and Wu [51] | Intention and behaviour toward open data publication | Government agency | TAM, institutional theory | Data providers |
| Wang and Lo [11] | Adoption decisions | Government agency | TOE | Data providers |
| Shkabatur and Peled [38] | Post-adoption | Country (The Philippines, Moldova, Kenya, Brazil, and Morocco) | DOI Theory | Data providers |
| Kaasenbrood, et al. [52] | Adoption decisions | Private organizations | N/A[a] | Data users |
| Maccani, et al. [37] | Pre-adoption, adoption decisions, post-adoption | Business company | DOI theory | Data providers |
| Susha, et al. [35] | Adoption decisions | Business company | Unified theory of acceptance and use of technology | Data users |
| Estermann [53] | Adoption decisions | Heritage institutions in Switzerland | DOI theory | Data providers |

[a]N/A: Not available.

that the IT innovation possesses and the adopter's perceived behavior, research on IT innovation adoption process has evolved implicitly through the body of knowledge from other disciplines. The next sub-sections describe the theoretical foundation of this study.

## 3.1 Innovation adoption process

The technologically oriented organizational innovation and adoption literature is based on the early paradigm of social change proposed by Lewin [58]. The process of change, according to Lewin's model, is essentially a sequence of three steps: i) unfreezing, ii) moving (or changing), and iii) refreezing. The "unfreezing" phase trains the system for transition, learning new behavior trends in the movement of the organization or unit or agency, and assimilating the consequences of change. The moving or changing step refers to shifting a group or unit to learn newly acquired trends, and the outcome of the transition is assimilated [58]. The

refreezing step allows certain behavioral habits to endure and become a lasting feature of the system [58].

In accordance with the evolution of the innovation adoption process, the innovation theorists have further discussed various stages of innovation adoption that lead to categorizing the innovation adoption process into multi-phases such as pre-adoption and post-adoption (continued usage) [59], initiation, adoption, and implementation [60], and initiation, adoption, implementation, and evaluation [61]. Studies by Kwon and Zmud [62] and Cooper and Zmud [63] suggested a six-stage IT implementation model, as presented in Table 2. Followingly, Damanpour and Schneider [64] suggested that the six-stage IT implementation model can be grouped into three broad phases, namely pre-adoption (initiation), adoption decisions (adoption, adaptation), and post-adoption (acceptance, routinization, infusion). The first phase is known as the pre-adoption or initiation phase, in which activities such as getting familiar with the innovation, planning to acquire the innovation, and proposing the innovation to be adopted take place [54,59]. The second adoption phase is the adoption (decision) phase, a mechanism that shifts from pre-adoption to defined adoption. In this phase, the personnel of an organization becomes conscious of an innovation and accesses knowledge that can help draw a decision on whether to accept or reject the innovation [59,64,65]. The third and final phase in the innovation adoption process is the post-adoption phase [59,64]. In the post-adoption phase, the adopter acquires the innovation and is set to be implemented or established until it becomes a regular feature in the adopter's environment [54,56,64]. This indicates that implementation is one of the post-adoption activities, while post-adoption is the phase that happens after an innovation has been decided to adopt during the adoption (decision) phase [54,57,64]. After a certain period, an innovation that is considered new at the time of adoption loses its identity as it has been embedded in the organization's task routine [54].

Following the innovation adoption process, this study resorted to using the post-adoption term to identify the consequences of OGD in data providers' environments. The question raised by [39], "*Why Haven't Consequences Been Studied More*?", suggested that post-adoption is a pro-found area of study. The fact that there are various desirable or undesirable effects on adopters or the social system, thus widens the post-adoption study perspective. The undesirable consequences that could happen in the post-adoption phase include decommissioning, stagnant, discontinuation of innovation, and the like that goes beyond pro-innovation bias [39].

This study focuses on the post-adoption phase of the innovation adoption process. The model from Cooper and Zmud [63] helped establish a more profound knowledge of IT and

**Table 2. Six-stage IT implementation model [63].**

| Stage | Description | Innovation adoption phase [64] |
|---|---|---|
| Initiation | The process of identifying the problems and the need for IT innovation as a solution. | Pre-adoption |
| Adoption | The decision-making process to execute an IT innovation and allocate the resources needed. | Adoption (decisions) |
| Adaptation | The IT innovation is created, installed, and retained, while IT innovation is learned to be used by the organization members. | |
| Acceptance | The stage during which the members of the organization are convinced to use and implement IT innovation. | Post-adoption |
| Routinization | The utilization of IT innovation is viewed in the organization as a daily practice. | |
| Infusion | IT creativity is integrated with the job structure of the organization. | |

operational problems throughout the implementation process. The model also forms the mechanism of IT-enabled organizational change and is relevant to the kind of IT innovation (OGD) and implementation background (public sector organization) in this research. Given that there is a relatively wide gap in OGD post-adoption studies, this study stands on the premise that OGD implementation is yet to be examined as a dependent variable in the post-adoption phase. With this notion, this study assessed whether the OGD implementation is progressing toward embedding the OGD in the public sector working system. Successful and beneficial adoption of an innovation is acknowledged when the innovation is put into practice and integrated into the organization [54,65,66].

### 3.2 Technology–Organization–Environment (TOE) **framework**

The theory of innovation adoption in an organization consists of a compilation of theories taken from several investigation fields that underlie much of the study of mechanisms of growth and technical progress [67]. Various studies have attempted to classify diverse influences as possible determinants of IT adoption in organizations. Wejnert [68] developed a system in which innovation adoption determinants were clustered into three key components: innovation characteristics, innovator characteristics, and environmental background. Similarly, Iacovou [69] structured the organizational readiness, benefits of the innovations, and external pressure in his framework for Electronic Data Interchange (EDI) in small businesses. However, Tornatzky, et al. [70] had the most recognized attempt to identify and categorize the determinants of technological innovation adoption in organizations in their book, *The Process of Technological Innovation*. Tornatzky, et al. [70] presented a framework for clustering the determinants of technological innovation adoption into three structural contexts that influence organizational innovation adoption and implementation decisions, namely the technology, organization, and environment (TOE). As a result, the TOE framework has become a prevalent theoretical perspective on IS/IT adoption [71]. This statement holds true in the OGD context, where a study conducted by Khurshid, et al. [72] has indicated that the TOE is the most utilized theory in OGD adoption research at the organizational level. In other studies, Hossain, et al. [5] posits that the TOE framework does not define explicit constructs making it adaptable across several IS/IT innovations thus receiving considerable attention in OGD adoption studies.

Additionally, the TOE framework addresses the requirement for more socioeconomic advancements and has received more substantial theoretical and empirical support in the IS domain [73] than many other adoption models [74]. According to Zhu, et al.'s [75] theoretical evaluation, the TOE framework is more important than Rogers's [39] Innovation Diffusion Theory (IDT). The TOE framework's technological and organizational environment mirrors that of IDT's own, however, the TOE framework is is deemed more adequate for explaining technology adoption because it also incorporates environmental aspect as additional constructs [76].

The technology context represents the availability and characteristics of a particular technology. The organization context includes structured and informal systems of connections, coordination methods, scale, and slack of the organization. Lastly, the environmental context consists of business features and market structure, resources for technological support, and government regulation [70]. It may also contain an agency external to the company with relevant experience to assist in IS/IT adoption [77]. Nevertheless, the ability to modify the variables or measurements within each construct renders the TOE configuration highly flexible in a wide variety of IS/IT innovation [5,78]. Scholars have also found no reason for the principle itself to be modified. As the TOE paradigm contains variables and has great analytical

influence in a broader sense, to expand and enrich theoretical lenses, Tornatzky, et al. [70] did not implement a fixed model and instead suggested combining the TOE framework with other theories [66,74,79]. Previous scholars have consistently focused on the variables within the TOE's framework to improve its theoretical ground and empirical compliance.

In brief, TOE presents a viable method for researching organizational IS/IT implementation through a range of technologies. Various forms of innovations have diverse influences that affect their adoption. Likewise, different socioeconomic, geopolitical, or cultural backgrounds would often have varying influences [78]. The TOE framework extends the debate on implementation beyond a technical narrative and combines organizational and external viewpoints. Nevertheless, the framework's technology, organization, and environment contexts raise both limitations and potential for innovation adoption in an organization [70].

The innovation adoption process theory provides advantages in understanding the dynamics of influences and adoption patterns in an organization. At the same time, the Technology-Organisation-Environment (TOE) shows equal strengths in a heterogeneity of use across organizations. However, the TOE framework has a low ability to explain the process of change (explanatory power) of innovation adoption. Considering the strengths and limitations of both models, the innovation adoption process and the TOE framework complement each other in presenting a comprehensive evaluation framework for OGD post-adoption in the Malaysian public sector.

## 4. Research model and hypothesis

OGD is regarded as new government innovation and can be conceptualized using the innovation adoption process [9]. Driven by the past literature, document analysis, and consultations with field experts, this study resorted to the TOE framework to model the OGD post-adoption framework attentively. Researchers have combined the DOI and innovation adoption process with different contextual frameworks to address OGD post-adoption in the public sector. This study employed a deductive approach in developing the theoretical framework. Deductive research is concerned with developing a hypothesis on existing theory and thus relies heavily on the theoretical framework to validate the hypothesis [80]. Although there is a less prescribed procedure for constructing a theoretical system proposed by deductive scholars, this study reflects most of the literature's general measures.

There are numerous factors that can influence OGD post-adoption, but this study has identified eight factors as the independent constructs of OGD post-adoption. These constructs were gathered during the preliminary study through a series of semi-structured interview sessions with OGD stakeholders and were consolidated according to the TOE framework. The independent constructs are compatibility, complexity, relative advantage, organizational culture, top management support, IT competency, data demand, and incentives. On the other hand, three dependent constructs (acceptance, routinization, and infusion) were derived from the post-adoption phase of the innovation adoption process theory. The research model of this study is presented in Fig 1.

### 4.1 Compatibility

Compatibility is described as the degree to which an invention is consistent with its business process, belief system, and culture [54,81]. Rogers [54] also stressed the importance of an innovation's compatibility with an individual's work responsibilities and value system in increasing the likelihood of its adoption. While Tornatzky and Klein [82] view compatibility as a multidimensional notion that includes normative or cognitive compatibility (i.e., how adopters perceive an innovation) as well as operational or practical compatibility (i.e., compatibility with

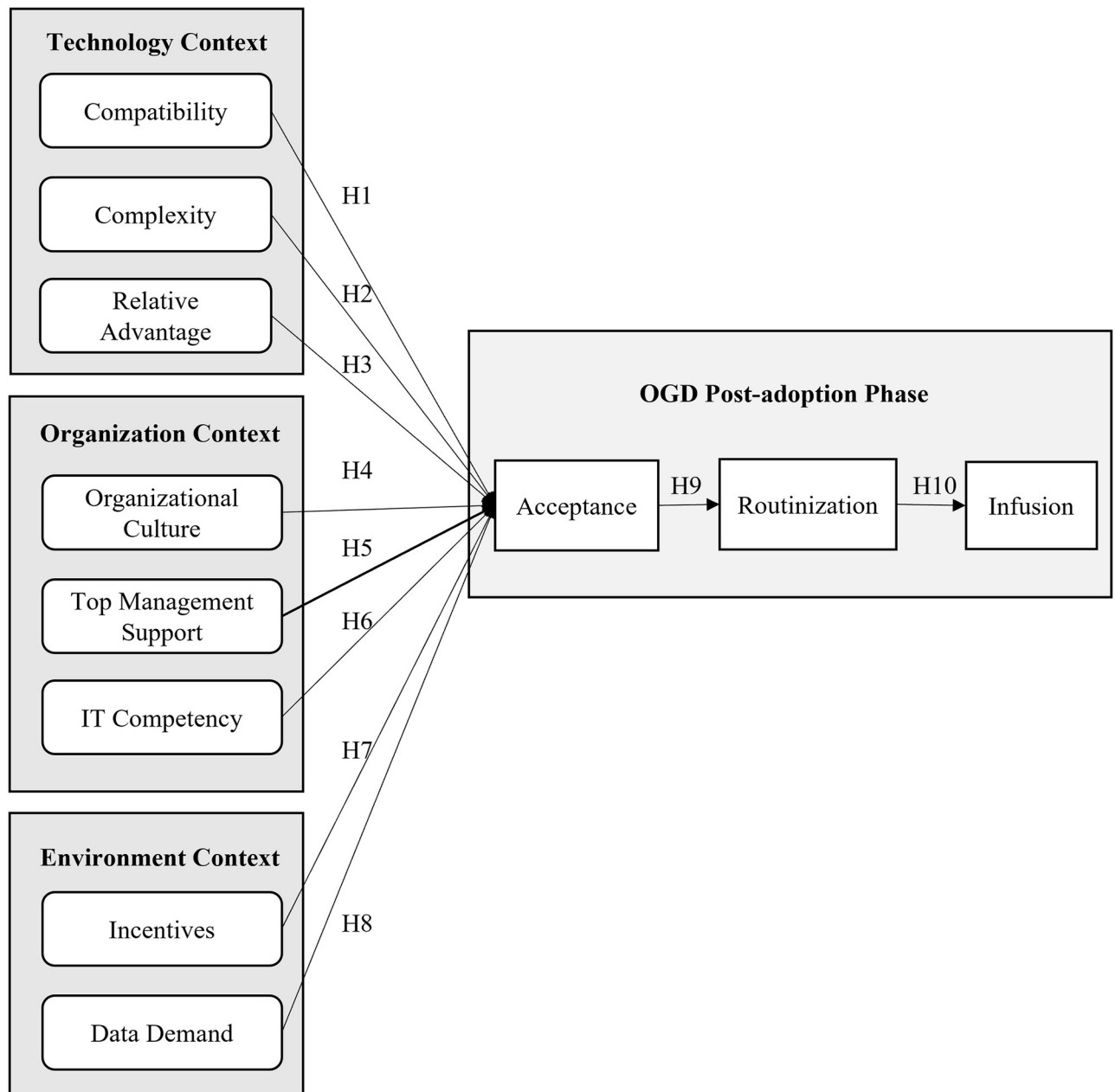

**Fig 1. The research model for the OGD post-adoption framework in the public sector.**

what adopters do). In this study, compatibility is viewed as whether the OGD is aligned with the government agency's norms in data management and sharing. For instance, Elixmann and Jarke [83] observed in their research that the OGD produces problems with the data management done by the public administration. Government agencies may lose authority and sovereignty over their data, which could lead to the data being incorrectly interpreted or utilized for harmful purposes. This relates to the idea that the person or organizational unit that collected the data is the exclusive owner of the information and has the right to use it in any way they see fit.

Moreover, government agencies do not have a culture that places a strong emphasis on errors; as a result, members of the administrative staff are motivated to take precautions to minimize the likelihood of making mistakes [83]. Additionally, the spirit of openness and disclosure that OGD promotes will prevail over the confidentiality practices employed by government agencies [83]. Government agencies perceived that the OGD policy is not aligned with the agency's business procedure as all data are treated as classified data. However, some of the created data are crucial and should be shared with the public, such as facts relating to consumerism, the environment, or criminal activity, to mention a few. The combined understanding of OGD compatibility by these data providers is thus a very important aspect to be further investigated. Hence, the next hypothesis is suggested:

*H1*: *Compatibility positively influences OGD post-adoption in the public sector.*

## 4.2 Complexity

Complexity may be referred to as the degree to which innovation is viewed as difficult to use and comprehend [54]. Innovations that are more challenging to implement are less likely to be taken up by organizations. Diffusion of complex innovations is common in organizations with low levels of expertise. As a result, the company is unable to fully benefit from the new innovation. In addition, the complexity of innovations results in increased resistance to change among users because of a lack of knowledge and skills on their part [84]. For example, in Kleiman, et al. [85], the civil servants perceived that data disclosure might be alarming if they believed that opening government is too complicated and convoluted or if they lack an understanding of the fundamental processes required to make data available to the public. In a similar vein, Elixmann and Jarke [83] discovered that the level of complexity of OGD implementation was one of the variables that contributed to the resistance to adhere to the open data policy in the German Public Administration. While the local study by Mustapa, et al. [86] highlighted the government agencies' perceived complexity of OGD because of the certain processes before the data can be released, thus, publishing OGD is viewed as tediously complicated. The process mentioned includes the data required to be in a high granularity, anonymous, and readily in a machine-readable format. Choosing the appropriate data as an OGD is already a complex challenge for certain data providers because of the fear of releasing incorrect data. Hence, the next hypothesis is proposed:

*H2*: *The complexity of OGD negatively influences OGD post-adoption in the public sector.*

## 4.3 Relative advantage

Relative advantage has been established as a major factor in multiple innovation adoption research at the organizational level [87]. By definition, the relative advantage is the degree to which innovation is considered beneficial and may offer advantages to the organization [39]. The relative advantage is generally expressed as the degree of perceived usefulness that innovation may bring to the organization, and thus, the relative advantage and perceived usefulness are used synonymously in the literature on innovation adoption [69,76]. Despite the fact that perceived usefulness and relative advantage seem to be pertinent to this study, researchers have pointed out their parallels [40,88]. Furthermore, Moore [88] contends that an innovation is often generated with the intention of serving a certain goal, and in order for it to be accepted, it must be viewed as serving this purpose to a greater degree than its predecessors did. As a result of its high generalizability, the idea of relative advantage carries with it an allure that is

reminiscent of common sense. Given the instinctive enticement of Roger's [39] concept and its frequent recurrence in the IT adoption's body of research, the term relative advantage is chosen in this study.

Many studies have identified the benefits of OGD to society; however, little is known about whether OGD provides the same benefits to data provider organizations. For example, in Yang and Wu [89], the researchers found that the government agencies perceived that the OGD would give an advantage to the agency's governance performance. With effective governance in place, the government agency would be able to prioritize which datasets can be made public and plan the next steps. The analysis by Yang and Wu [89] also revealed that the government agencies were concerned about whether the published datasets would attract the public to use them because unused datasets undermine the government agencies' commitment to OGD. In order to find the equilibrium, it will therefore be anticipated that data providers perceived OGD post-adoption would provide benefits to the organization in the long term. Thus, the next hypothesis is formulated:

*H3*: *Relative advantage of OGD positively influences OGD post-adoption in the public sector.*

### 4.4 Organizational culture

Organizational culture can be defined as a group of individuals who have the same convictions and awareness of challenges inside and beyond the organization [90]. The organizational culture in this research refers to the openness culture of data providers, which focuses on how they react to managing and sharing data openly. As Ke and Wei [91] described, the relationship between organizational culture and IS acceptance is extremely vital as it can lead to either resistance or modification of the IS to suit the organization's culture. The organizational culture and OGD have a strong relationship as the former directly influences OGD publication [46]. Studies by Ruijer and Huff [92] and Yang and Wu [12] suggested that the government's openness culture would be easier to encourage once the government agencies are prepared to exchange knowledge and data outside the agency itself. Additionally, Zhao and Fan [10] considered organizational culture an intangible asset because it may affect an agency's awareness. In particular, scholars opine that a culture of openness should be fostered for open data to become the default standard for working procedures within the public sector [12,93]. However, there are cases where data providers seemed to be receptive to OGD but implicitly disregarded the initiative owing to the long-standing risk-avoidance culture instilled in the organization [2,94,95]. Study by Zhao, et al. [96] also denotes that organizational culture affect OGD implementation which stunted OGD performance in China. Such a scenario triggers the organizational culture to be investigated in the OGD post-adoption phase. Hence, the following hypothesis is posited:

*H4*: *Organizational culture positively influences OGD post-adoption in the public sector.*

### 4.5 Top management support

A number of studies have shown that top management support is one of the most critical factors contributing to the effective introduction of IS/IT innovation in all adoption phases [97,98]. The top management support factor is essential in OGD post-adoption for three main reasons. First, top management support is vital to increase the effort and resources to build an atmosphere more favorable for the adoption of OGD [99,100]. A lack of resources is a common issue for the deployment of any innovation in the government agency, and this is where

the top management roles are crucial for making important decisions on allocating the necessary resources [77]. Second, top management has the authority to circumvent subordinate resistance to OGD initiatives during the change management process. Third, top management is responsible for constructing an effective communication mechanism in promoting innovation adoption in the organization [78]. Weak coordination regarding the strategic advantages of innovation also contribute to resistance within the stakeholders of the organization to consider the adoption of innovation [101]. Additionally, the management level has the authority to decide which data sets can be published [5]. A study Wang and Lo [11] explored the top management factor as the most influential factor in the OGD adoption phase. Similarly, Hossain, et al. [5] examined management leadership as one of the positive factors in OGD initiative performance. It is essential to investigate if this factor continues to provide the same contribution in the post-adoption of OGD. The following hypothesis is thus suggested:

*H5*: *Top management support positively influences OGD post-adoption in the public sector.*

## 4.6 Information technology competency

Information technology (IT) competency construct is an integrative concept that combines the organization's human capital and physical resources [102]. In this regard, the human capital dimension describes the organization's members' experience, skills, and knowledge [97]. Simultaneously, the physical resources refer to the IT infrastructure used to implement OGD, such as a personal computer, servers, internet connection, and whatnot. The human capital and physical resource dimensions complement each other to portray the organization's capacity to adopt innovation. Other terms used to reflect the same meaning of IT competency that has been adopted by other studies are IT expertise [103], technical competence [81,101], technical capacity [10], technological competence [104], and IT sophistication [98]. Previous research suggests that IT competency which is normally related to the infrastructure and human capability seems to be a significant factor in the adoption of innovation in an organization [97,105].

Handling OGD initiatives need a substantial amount of data processing effort, starting from collecting, cleaning, harmonizing, and formatting before the data can be published [5]. Hence, it is essential for the government staff in post-adoption of OGD to be IT competent to continuously support OGD. The IT competency factor has been highlighted as part of the key characteristics of a successful OGD implementation [5,25]. Furthermore, as posited by Janssen, et al. [2], insufficient knowledge and skills and unsuitable data infrastructure can cause many datasets to continue to be out of sight. The IT competency in this study aims to assess the government agency's capability in IT infrastructure readiness and the technical skills, experiences, and knowledge of data providers personnel in running OGD initiatives. In accordance with the foregoing, the following hypothesis is proposed:

*H6*: *IT competency of the government staff positively influences OGD post-adoption in the public sector.*

## 4.7 Data demand

Data demand is a new element learned during the meeting with government agency representatives. Data demand carries the meaning of data requests by the public for particular datasets that are related to the organization. Provided that the data are non-confidential, the public can request datasets through the government data portal by submitting a data request form. If a data

consumer already knows which government agency provides the data they want, an email for data request is sent directly to the government agency. A study from [89], observed that data demand did play the role as the external driver for government agencies to publish data. In their study, Yang and Wu [89] assert that government agencies received pressure from various parties, including academia, non-profit organizations, and the public on social media, requesting the government to release data. Some social communities even put forward innovative ideas for using open data, which encourages government organizations to release more datasets [89]. This is corroborated by a study by Li and Chen [106], who discovered that the lack of external pressure on government bodies demotivated them from pursuing OGD initiatives. However, the OGD ecosystem in Malaysia's public sector is highly supply-driven [107]. Undoubtedly, meeting the data demand from the interested parties can be quite challenging. Although occasionally, the requested data are categorized as open, the process of aligning the data to the OGD requirements is tiresome, thus hampering the government agency in fulfilling the request. However, a different narrative is manifested if data providers feel that releasing their data will benefit them. The data demand is seen as having a favorable impact on the post-adoption of OGD, but this is not yet supported by empirical studies. Consequently, the following hypothesis is proposed:

H7: *Demand for OGD from the public positively influences OGD post-adoption in the public sector.*

## 4.8 Incentives

Incentives can be defined as any kind of recognition provided to a person or social system to encourage an explicit shift in behavior [39,108]. The "incentives" factor has been identified by Rogers [39] and Kulkarni, et al. [108] as a positive antecedent on influencing the adoption of innovation in the organization and increasing the rate of innovation adoption. In one of the findings by Rogers [39], providing incentives was also one of the organization's strategies to secure adoption at a specific rate, but once the desired adoption rate was achieved, the incentives were discontinued. In addition, Kulkarni, et al. [108] indicated that incentives or any nonmonetary rewards are essential to encourage the successful implementation of knowledge management in an organization.

While in the OGD area, Shkabatur and Peled [38] asserted that a lack of incentives has led to a poor institutionalization of OGD in four developing countries, namely, the Philippines, Morocco, Moldova, and Kenya. The findings convey the impression that when prizes such as cash money, national recognition, or awards are offered as an appreciation for their involvement, government agencies appear to be more driven to OGD. The same sentiment was expressed by Devriendt, et al. [109] and Elixmann and Jarke [83], who suggested that the data owners were discouraged from sharing their data due to limited rewards. A study by Zhang, et al. [110] further underlined the importance of incentives, claiming that rewards for government officials may help to reduce resistance to OGD.

In this study, the incentives factor may come from the organization's internal or external environment. The incentives include monetary or non-monetary rewards such as training opportunities, compliments, or recognition from the organization's management to bring about a desired behavioral shift [39,111]. Furthermore, the involvement of government agencies in OGD initiatives was mostly voluntary; therefore, acknowledging their contributions is justified [112]. The incentives factor aims to investigate whether the incentives positively influence the post-adoption of OGD. Therefore, the following hypothesis is formulated:

H8: *Incentives positively influence OGD post-adoption in the public sector.*

## 4.9 Acceptance

Acceptance is defined as the effort to bring the organization's members to use or practice an IT/IS innovation [113]. In this study, OGD adoption is a decision made by the federal government system from the higher management level; hence, the OGD initiative is eventually expected to be adopted by government agencies at all levels. The OGD acceptance implies that the government agencies must be convinced to commit to OGD by considering the influence factors of organization, technology, and environmental characteristics. Numerous studies were discovered examining OGD acceptance from the OGD users' perspective among various user groups such as academics, the public, journalists, app developers, and civil servants [30,32,33,114]. However, study on OGD acceptance among government agencies in the post-adoption phase is limited. The acceptance of OGD among government agencies in the post-adoption phase is important because it determines how OGD can be induced in the government agencies' work norms.

Many researchers thought the adoption of a particular innovation would result in nothing but positive outcomes for its adopters. According to Rogers [39], the mentioned scenario is called pro-innovation bias. The truth is that not all innovation adoption in an organization reaches the desirable state [39]. This situation is perceptible for OGD as a handful of studies denote the resistance to OGD adoption from the data providers' perspective [23,83,106]. At the same time, the barriers to OGD implementation by various government agencies [17,115,116] reflect that OGD has not fully been accepted by the data providers. The acceptability of OGD policies is significant as it defines the conduct of the government agencies in the next post-adoption phase. OGD acceptance within organizations is crucial as the desired results cannot be realized without functional support and commitment from the data providers [17]. Therefore, the following hypothesis is formulated:

*H9*: *OGD acceptance positively influences OGD routinization in the public sector.*

## 4.10 Routinization

Routinization is referred to as the extent to which the innovation has been established and become part of the organization's work systems [39,63]. Routinization is pursued after the acceptance stage in the post-adoption phase, which implies that the OGD cannot be in the normal practice of the public sector task function if it is not well received. At this stage, the innovation will also lose its identity and become a regular activity in the organization [39]. Some of the activities in routinization stage includes publishing OGD in a regular basis and has become a normal operation in government agency. Numerous obstacles can delay or halt the implementation of an innovation within an organization, and one of them is starting the innovation with overly large aims [39]. An organization that rushes to complete the implementation of an innovation may overlook critical stages in the innovation adoption process [39]. For instance, if the acceptance stage is rushed, the routinization stage might never occur because of the potential difficulties with the acceptance stage that might ensue. Nevertheless, in order to avoid a significant risk, the innovation adoption process must be implemented sequentially [39].

To the author's knowledge, there has not yet been a study on the routinization of OGD from the point of view of data providers. A study by Dawes, et al. [8] has exhibited a strong commitment to OGD publication as a significant factor for a well-developed OGD ecosystem in the city of New York. Another work by Zuiderwijk, et al. [117] discussed and associated the consistent data dissemination practices in day-to-day activities as a success factor in open data implementation, but the study does not profoundly investigate routinizing OGD publication. By adhering to routinization activities, the publishing process of OGD across departments or

agencies may be standardized, giving each government agency a consistent baseline to infuse OGD in their work's norm. Hence, the next hypothesis is suggested:

*H10*: *OGD routinization positively influences OGD infusion in the public sector.*

### 4.11 Infusion

Infusion is the process of embedding IS/IT into an adopter's work system and leveraging its complete and fullest potential [118]. A few researchers have employed infusion as the highest degree stage in the implementation of IS/IT at the individual or organizational level. The majority of the studies in the literature have investigated infusion at the individual level [13,119,120], while only a few focused on infusion at the organizational level [16,19,121]. At an individual level, the infusion stage is keen to explore the extent of individuals' use of technological innovation. In contrast, the infusion stage at the organizational level investigated how the IS/IT can be fully leveraged in order to gain benefits and positive impact. The infusion stage of IS/IT at an organizational level is crucial to ensure the long-term use and sustainability of the IS/IT. In addition, the infusion stage would prevent the IS/IT from becoming underutilized and obsolete. Furthermore, a number of studies accord that underutilization of IS/IT beyond adoption has been identified as one of the factors in organization failure to realize the full value of their IT investment [16,81,122].

In the context of this study, the infusion of OGD is viewed as embedding the OGD publication in the data providers' work system. The OGD infusion is intended to ensure that OGD publishing occurs on a continual basis rather than as a one-time activity for data providers. This is because OGD must be sustained over an extended period of time in order to gain benefits and generate more value for the citizen [20,21]. Moreover, Zuiderwijk and de Reuver [17] found that lack of sustainability appears as the new concern to achieving OGD's intended objective. However, no studies on the infusion of OGD from the standpoint of data providers have been conducted. Thus, this study views infusion as a crucial stage in OGD post-adoption toward sustainable and long-term implementation of OGD among data providers.

### 5. Methodology

The research framework for this study is dissected in two main phases. The first phase is called "exploration," in which the initial factors of OGD in the post-adoption phase were gathered in a preliminary study. The second phase is referred to as the "development" phase, where the researcher develops the OGD post-adoption framework and the research instrument. The main works of the study were performed throughout the second phase. Fig 2 illustrates the research framework for this study.

During the exploration phase, a preliminary study was conducted by employing a semi-structured interview method with three high-ranked government officials from the central agency as the interview participants. Each of the interview sessions lasted about one hour and was voice-recorded for transcription. The interview participants were given a consent form and subject information sheet prior to the interview session. The participant was deemed to agree to give consent for the researcher to collect their personal data (Name, designation, signature, and date of signature) once the consent form was signed. In the subject information sheet, the participant was given a brief overview of the study, including an introduction and purpose of the study, study procedure, participant's right in the study, and confidentiality of the participant's details and answer. There was no privacy laws infringement as all personal details of the participants were not published elsewhere. The interview transcriptions were later analyzed using thematic analysis to extract meaningful information from each

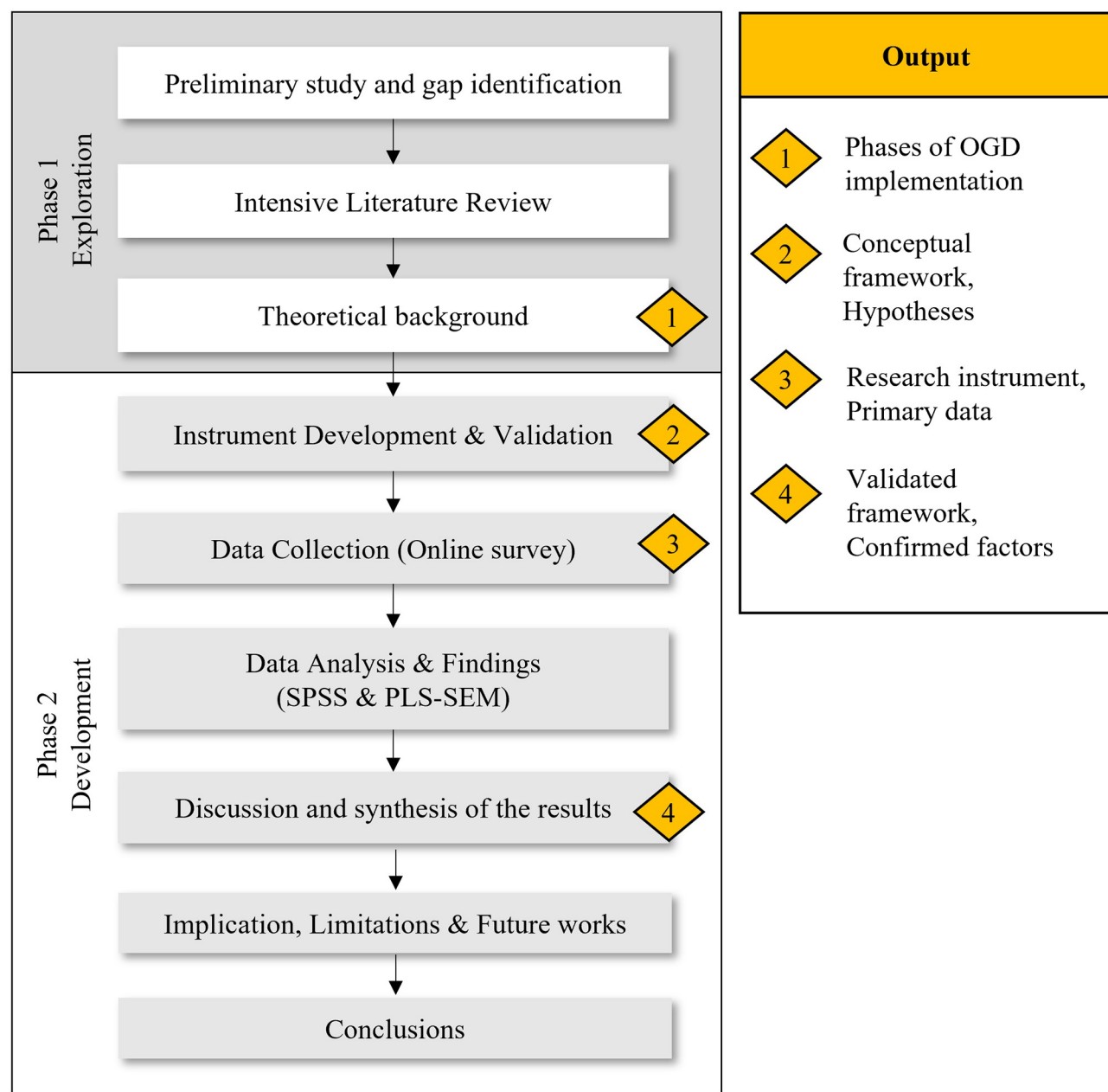

**Fig 2. Research framework (Source: Own illustration).**

interviewee. The semi-structured interview goals were to explore the current state of OGD implementation and gather the initial factors of OGD implementation in Malaysia's public sector. The output of this phase is the confirmation phase of OGD implementation in the Malaysian public sector and the initial factors that influence OGD post-adoption in Malaysia's public sector. The interview scripts can be referred at S2 Appendix.

In the second phase, the critical work included designing two essential components of this study, namely, the research model and the research instrument. The former is essential to provide a descriptive representation of the theory use and the relationship between variables in the research model. It also provides a context for examining a problem or phenomenon,

therefore constructing the rationale for developing the hypothesis [80]. Some scholars postulated that it is essential to combine more than one theoretical model to comprehend the phenomenon of IT adoption from a different viewpoint [76,123]. In addition, innovation theorists have suggested that the creation of a unifying innovation theory might not be feasible owing to the intrinsic variations across innovation types [124]. The innovation adoption in organization research has thus integrated adoption and implementation theories with frameworks from various contexts to assess innovation's adoption.

In the second phase, development, the research instrument was developed. The measurement items or the constructs were gathered during the preliminary study in the exploration phase. The constructs were compared with previous studies that used the same TOE framework in investigating innovation adoption in an organization. A content validity procedure was performed with eight panels to ensure the validity of the research instrument. However, given its lengthy steps and results, the content validity procedure will not be explained in this paper. Data collection was performed after the research instrument was refined according to the content validity results. Following the data cleaning procedure, the partial least square-structure equation modeling (PLS-SEM) analysis was conducted using SmartPLS 3.0 [125] software.

## 5.1 Context of study

The main subjects of this study can be defined as the government agencies in the Malaysian public sector that have implemented OGD initiatives. The government of Malaysia adopted the OGD program in 2014 and has been actively pursuing it ever since. A central agency called the Malaysia Administration Modernization, and Management Planning Unit (MAMPU) is authorized by the federal government's high-level management to spearhead the country's OGD implementation. MAMPU is responsible for supervising the country's policies and strategizing the OGD implementation nationally and globally. Subsequently, this mandate provides a clear path for the central agency to deploy OGD initiatives at the government agency from all levels. Among the task of MAMPU in OGD implementation is hosting the government open data web portal that can be accessed at www.data.gov.my. During the early phase of OGD adoption, less than two hundred datasets were published in the government open data portal. Some of the datasets were not in a machine-readable format, and to add to the setback, they could hardly be re-used using computer applications. As of today, there are more than 12,000 datasets from 18 different clusters have published their datasets. However, in recent years the publication rate of datasets in the government open data portal has dropped. This situation certainly raises questions about the direction of OGD in the post-adoption phase in Malaysia.

The population of the study is defined as the government agencies that play the role of data providers and are registered in the government open data portal, data.gov.my. Hence, simple random sampling was employed as the sampling technique for this study. In the simple random sampling technique, each of the population members has an equal chance of being selected to ensure that the sample is representative of the population [126]. The reason for choosing a simple random sampling technique was due to every single agency in the defined population having an equal probability of being chosen as the respondent. As a result, a government agency that has already adopted OGD was deemed appropriate as the respondent. A total of 671 data providers from all types of government agencies were listed as the user and furnished with full names and email addresses. The government data portal users were the respondent for this study as the person responsible for OGD implementation in their respective agencies. The representative should ideally be a senior officer such as a Chief Information Officer (CIO), Manager or Head of IT units, or a member of the IT Steering/Implementation committee or appointed open data champion or open data agent or person in charge that possess the

knowledge of their agency's practice and routine of OGD initiatives. Some agencies appoint an open data champion to be the corresponding personnel for matters pertaining to OGD initiatives. The type of government agency involved in this population includes the federal and state government, local authorities, statutory bodies from federal and state levels, and government link companies. According to Taherdoost [127], for the population around 600 to 700, the estimated sample size is 234 to 248, with a 95% confidence level and a 5% margin of error.

## 5.2 Data collection

Prior to the collection of primary data, the research instrument completed a content validity procedure and a pilot study. However, the content validity procedure and the pilot study scope were not included in this paper. The online research instrument was developed in a paid survey tool called Survey Monkey. The online research instrument was built in two language versions (Malay and English) to allow the respondents to choose the language they prefer the most.

Owing to the widespread use of the internet in everyday life, an online survey is the most convenient method for respondents to reply at their convenient location and time [128]. Conversely, the online survey enables the researcher to gather responses with less effort and more immediately. Furthermore, being a user of the government data portal requires the respondents to be computer-literate and use the internet most of the time at work. Therefore, there should be a minimal concern for the respondent to volunteer to answer the online survey. The respondents were invited to respond to the online survey through an email invitation. The respondents' consent to participate in the online survey was obtained once the respondent answered the survey. The respondents were allowed to ignore the email invitation if they did not agree to participate. The respondents were also permitted to abandon the survey if they did not feel comfortable answering all questions. Personal details that were captured in the online survey form include respondents' position in the agency, job title, gender, age, and service term. The respondents' information was not used and shared for any other purpose except for this study; therefore, this study met the personal data and privacy laws. A cover letter was attached to explain the purpose of the study. The questionnaire of this study was developed in the close-ended type of questions and self-administered by the respondents. The final item measurement is shown in Table 3. The complete survey questionnaire is presented in the S1 Appendix.

Data collection was conducted for a period of two months. After the first four weeks, the first email reminder was sent to the respondents who had partially or had not been totally responsive to the survey. At the end of the survey period, a total of 294 (44%) government agencies responded to the online survey. However, after a data screening process, a total of 28 respondents did not complete the survey, making the total number of data that can be considered complete and eligible for further analysis to 266, or equal to 40% of the total number of the targeted government agency. Therefore, a valid N number for this study is 266 ($N = 266$). According to Sekaran and Bougie [80], the rules of thumb from Roscoe, et al. [130] can be applied to determine the appropriate sample size for a population under study. The researchers advocate the minimum sample size of 30% for a population larger than 30 and less than 500. Thus, the study hit a minimum of 40% of the population for a sample size of 251. This result justifies the rationale to perform the data analysis as the following procedure.

## 6. Data analysis and results

The primary data analysis was conducted by applying the Partial Least Square-Structured Equation Modeling (PLS-SEM) analysis technique using SMART-PLS software [125]. The PLS-SEM techniques are known as second-generation data analysis techniques that allow for

**Table 3. Variables and final measurement items.**

| Constructs | Items' Code | Items |
|---|---|---|
| Compatibility (Adapted from Zhao and Fan [10], Junior, et al. [129]) | CPB1 | Open Government Data initiatives are compatible with the data captured at our agency. |
| | CPB2 | Open Government Data initiatives are suited to our agency's existing operating practices. |
| | CPB3 | Open Government Data initiatives are compatible with our agency's IT infrastructure. |
| | CPB4 | Open Government Data initiative is consistent with our agency's values and beliefs. |
| Complexity (Adapted from Yang and Wu [12], Junior, et al. [129]) | CPX1 | Our agency finds publishing Open Government Data is a complex process |
| | CPX2 | Our agency faces difficulty in categorizing data that can be published as Open Government Data. |
| | CPX3 | Our agency finds it is challenging to publish Open Government Data with high granularity. |
| | CPX4 | Our agency's data need to go through a complex process before being published as Open Government Data. |
| Data Demand (Self-developed) | DAD1 | Our agency regards the data request from the public as part of the government service to the people. |
| | DAD2 | Our agency believes that the use of open data from the public will influence our agency to publish Open Government Data. |
| | DAD3 | Our agency only accepts data requests for the already available datasets to be published. |
| | DAD4 | Our agency finds that fulfilling the demand for Open Government Data by the public is a satisfying task |
| Incentives (Self-developed and adapted from Wang and Lo [11]) | INC1 | The superior level agency provides our agency with incentives to implement Open Government Data initiatives. |
| | INC2 | An incentive from the superior level agency is essential for the agency at the bottom level to implement Open Government Data initiatives. |
| | INC3 | Our agency is more motivated to implement Open Government Data initiatives if the incentive is provided. |
| | INC4 | There is recognition provided by an external party (non-governmental/private bodies, etc.) for government agencies implementing the Open Government Data initiative. |
| Organizational Culture (Adapted from Yang and Wu [12]) | OGC1 | Our agency is willing to share information and data with the public. |
| | OGC2 | Our agency encourages the practice of information and data sharing with the public. |
| | OGC3 | Our agency is open to innovative policies such as sharing information and data with the public. |
| | OGC4 | Our agency has implemented the open government data sharing policy. |
| IT Competency (Adapted from Yang and Wu [12], Zhao and Fan [10]) | ITC1 | Our agency is committed to assuring that the staff is familiar with Open Government Data initiatives. |
| | ITC2 | Our agency has a sound knowledge of Open Government Data initiatives. |
| | ITC3 | Our agency has the technological resources to manage Open Government Data implementation. |
| | ITC4 | Our agency's staff is able to use their experience and knowledge to operate Open Government Data implementation. |
| Relative Advantage (Adapted from Yang and Wu [12]) | RAD1 | Open Government Data implementation increases the performance of our agency's operation. |
| | RAD2 | Open Government Data implementation raises the efficiency of our agency's operation. |
| | RAD3 | Open Government Data implementation enhances the effectiveness of our agency's operation. |
| | RAD4 | Open Government Data provides our agency with valuable information to make decisions. |
| Top Management Support (Adapted from Wang and Lo [45], Wang and Lo [11]) | TMS1 | Top management in our agency is articulating a vision for Open Government Data implementation. |
| | TMS2 | Top management in our agency is formulating a strategy for Open Government Data implementation. |
| | TMS3 | Top management in our agency is deploying the Open Government Data initiative implementation efforts. |
| | TMS4 | Top management in our agency is giving attention to the performance of the Open Government Data initiatives implementation. |
| Acceptance (Adapted from Yang and Wu [12]) | ACC1 | Our agency publishes Open Government Data voluntarily. |
| | ACC2 | Open government data initiative is well accepted in our agency. |
| | ACC3 | Our agency is satisfied with the Open Government Data implementation in our agency. |
| | ACC4 | Our agency published Open Government Data as frequently as possible. |
| Routinization (Self-developed) | ROU1 | Our agency publishes Open Government Data on a regular basis. |
| | ROU2 | Open Government Data publication has become a normal operation in our agency. |
| | ROU3 | Open Government Data publication is regarded as a regular activity in our agency. |
| | ROU4 | Our agency's work system is adapted well to Open Government Data initiatives. |

(*Continued*)

**Table 3.** (Continued)

| Constructs | Items' Code | Items |
|---|---|---|
| Infusion (Self-developed) | INF1 | The Open Government Data policy has been fully adopted by our agency. |
| | INF2 | Our agency has incorporated the publication of Open Government Data into agency work norms. |
| | INF3 | Our agency has implemented the Open Government Data initiatives in accordance with the guidelines set out. |
| | INF4 | Open Government Data publication is an integral part of our agency's activity. |

modeling and testing of the relationship among multiple independent and dependent constructs, all at once [131]. Researchers who employ PLS-SEM analysis typically present their visual hypotheses and variables in a path model diagram [132]. The PLS-SEM path model is formed by two sub-models, namely, the measurement and structural models. The next sub-section explains both models.

## 6.1 Measurement model assessment

The measurement model or the outer model represents the relationship between latent variables (unobserved variables) and the indicators (observable variables). The measurement model specifies how the latent variables are measured. There are two different ways to measure latent variables, i.e., reflective or formative measurement. In the former, paths connecting constructs to indicators are directed toward the indicators [131]. The associations between the reflective construct and measured indicator variables are referred to as outer loadings [133]. In the formative constructs, paths connecting constructs to indicators are directed toward the constructs. The relationship between the formative constructs and measured indicator variables is referred to as weights [133].

Although the specification of constructs is based on construct conceptualization and the study's objective, they are not always reflective or formative [134]. Instead, researchers may supply the theoretical reasoning with an empirical test called Confirmatory Tetrad Analysis (CTA) to determine the mode of the measurement model [135]. In order to use CTA, the constructs must at least have four indicators; otherwise, the CTA would not be evaluated [132]. This study assessed the CTA using the SmartPLS [125] software. Constructs with a significant $p$-value ($p < 0.05$) indicate that the construct should be modeled as formative, while constructs with a non-significant $p$-value suggest that the construct should be modeled as reflective. Table 4 provides the result of CTA for the measurement model.

As shown in Table 4, four constructs (Complexity, Data demand, IT Competency, and Routinization) have a significant $p$-value ($p < 0.05$), and there was a 0 value straddle between the confidence interval low adjustment and up adjustment [132]. However, given that this is the earliest study that explores the constructs specifically for OGD post-adoption, all constructs were modeled as reflective. In order to model constructs as formative, a thorough set of indicators that accurately reflect the construct's domain is required and must be supported by prior research [131,132]. A crucial feature of formative indicators is that they cannot be substituted for one another, which is not the case for reflective indicators [132]. Important components of the structure would be omitted if the preceding observations were disregarded [131]. Fig 3 illustrates the constructs modeling in SmartPLS software. This study presents four different types of measurement model assessments in the following sub-section.

**6.1.1 Internal consistency.** The internal reliability consistency measures whether all the indicators of a construct measure the same element. There are two tests for the internal

**Table 4. Confirmatory Tetrad Analysis (CTA) result.**

| No. | Constructs | *P* Values | Confidence Interval (CI) Low Adjustment | Confidence Interval (CI) Up Adjustment |
|---|---|---|---|---|
| 1. | CPB1,CPB2,CPB3,CPB4 | 0.220 | -0.055 | 0.016 |
| 2. | CPX1,CPX2,CPX4,CPX3 | 0.002* | 0.021 | 0.115 |
| 3. | DAD1,DAD2,DAD4,DAD3 | 0.000* | 0.049 | 0.149 |
| 4. | ITC1,ITC2,ITC4,ITC3 | 0.017* | -0.082 | -0.003 |
| 5. | INC1,INC2,INC3,INC4 | 0.199 | -0.011 | 0.041 |
| 6. | CUL1,CUL2,CUL4,CUL3 | 0.542 | -0.018 | 0.032 |
| 7. | RAD1,RAD2,RAD3,RAD4 | 0.106 | -0.009 | 0.058 |
| 8. | TMS1,TMS2,TMS3,TMS4 | 0.375 | -0.015 | 0.036 |
| 9. | ACC1,ACC2,ACC3,ACC4 | 0.240 | -0.025 | 0.082 |
| 10. | ROU1,ROU2,ROU3,ROU4 | 0.006* | 0.008 | 0.078 |
| 11. | INF1,INF2,INF3,INF4 | 0.073 | -0.009 | 0.076 |

Note:

*$p < .05$.

consistency reliability test: Cronbach's alpha and composite reliability (CR). Numerous studies have traditionally used Cronbach's alpha (α) to measure the data's internal consistency and reliability. However, many scholars suggest refraining from using Cronbach's alpha test as it tends to provide a conservative measurement [136]. The alternative measurement of internal consistency reliability to Cronbach's alpha is the CR test [137]. This is because Cronbach's alpha's limitations underestimate the true reliability of the constructs. The acceptable value of CR must be 0.7 or higher [137]. The result from Table 5 has indicated that the internal consistency reliability of all reflective latent variables has been established.

**6.1.2 Indicator reliability.** The indicator reliability refers to the size of the outer loading and demonstrates the proportion of indicator variance that is explained by the latent variable [138]. The aim of the indicator reliability test is to evaluate how an indicator or group of

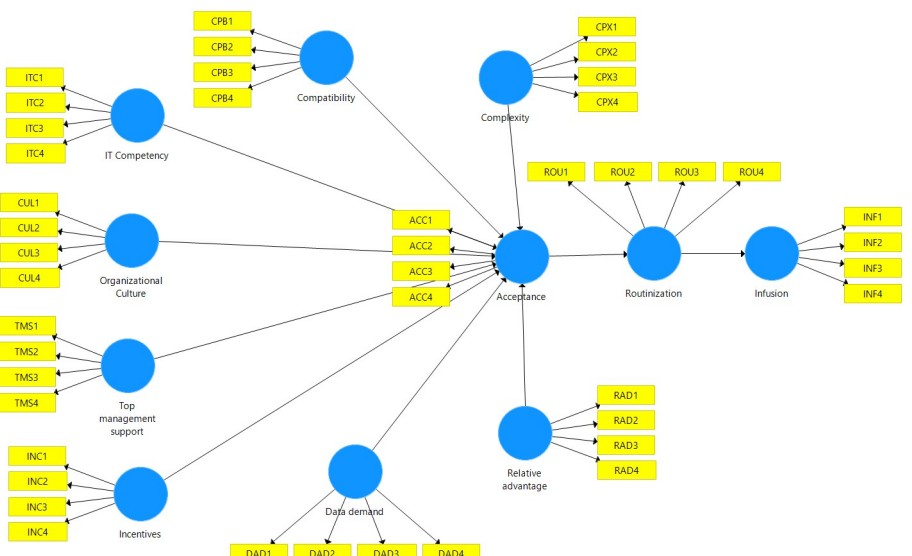

**Fig 3. The formative and reflective constructs modeling in SmartPLS software.**

**Table 5. Composite reality index and cronbach's alpha in internal consistency assessment.**

| Constructs | Composite Reliability | Cronbach's Alpha |
|---|---|---|
| Compatibility | 0.848 | 0.848 |
| Complexity | 0.881 | 0.820 |
| Data Demand | 0.893 | 0.820 |
| Incentives | 0.846 | 0.760 |
| Organizational culture | 0.899 | 0.831 |
| IT Competency | 0.855 | 0.801 |
| Relative Advantage | 0.899 | 0.851 |
| Top Management Support | 0.891 | 0.837 |
| Acceptance | 0.886 | 0.828 |
| Routinization | 0.895 | 0.842 |
| Infusion | 0.896 | 0.845 |

indicators is compatible with what is being measured [139]. As a basic rule of thumb, the uniform outer loading for an indicator should be 0.708 or higher to be retained in the PLS model [132]. The reason for the standard value of 0.708 is that the square of the indicator's outer loading should contribute to the variance of the latent variable by at least 50%. Calculation of a squared 0.708 will result in 0.50 (50%), and for that reason, the outer loading should be higher than 0.708 to get more significant reliability. On the other hand, indicators are suggested to be eliminated from measurements models if their outer loadings within the PLS model are smaller than 0.4 [132,140]. The final iteration results for indicator reliability for this study are encapsulated in Table 5. As part of the measurement model evaluation, two items, DAD3 and OGC4, were removed after two iterations of the PLS algorithm analysis owing to low factor loadings ($< 0.7$). The rationale behind the retained items is to ensure that the average variance extracted (AVE) value exceeds the threshold value of 0.5. According to Hair Jr, et al. [132], although the rule of thumb suggests removing an item lower than the threshold value, nonetheless, if removing an item will cause the AVE value to drop below 0.5, then the item can be retained. Removing too many items should be followed by some consideration as it could also trigger some effects on the content validity of the constructs [132]. The results for indicator reliability assessment are presented in Table 6.

**6.1.3 Convergent validity.** Convergent validity quantifies the extent to which an indicator is positively correlated with other indicators from the same construct [139]. In updated works by Hair, et al. [134], the convergent validity is known as "average variance extracted," which refers to the mean of all squared loadings from all indicators or items from a construct. In other words, AVE signifies whether indicators that are supposed to measure the same thing are highly correlated. Convergent validity is a necessary component of developing a valid construct. It is helpful in determining or finding the intensity of the relationship between two measures. One general thumb rule recommended that a latent variable should illustrate at least 50% of the variance of each indicator [131,132]. The rule means that the outer loading of an item from each construct should be above 0.708 as the squared number (0.7082) is equal to 0.50. The results for convergent validity and the factor loadings for each construct are presented in Table 7. All constructs achieved convergent validity with AVE above 0.5 after the low factor loading was removed in the indicator reliability assessment.

**6.1.4 Discriminant validity.** Discriminant validity is an essential assessment as it examines whether the investigated constructs are genuinely distinct from one another [131]. There are three forms of standards techniques to determine discriminant validity, namely i) cross-

**Table 6. Indicator loading index in indicator reliability assessment.**

| Constructs | Items | Loadings > 0.7 |
|---|---|---|
| Compatibility | CPB1 | 0.850 |
| | CPB2 | 0.889 |
| | CPB3 | 0.772 |
| | CPB4 | 0.803 |
| Complexity | CPX1 | 0.829 |
| | CPX2 | 0.812 |
| | CPX3 | 0.857 |
| | CPX4 | 0.714 |
| Data demand | DAD1 | 0.905 |
| | DAD2 | 0.903 |
| | DAD4 | 0.760 |
| Incentive | INC1 | 0.734 |
| | INC2 | 0.766 |
| | INC3 | 0.817 |
| | INC4 | 0.724 |
| Infusion | INF1 | 0.746 |
| | INF2 | 0.881 |
| | INF3 | 0.836 |
| | INF4 | 0.841 |
| Organizational culture | OGC1 | 0.869 |
| | OGC2 | 0.857 |
| | OGC3 | 0.867 |
| IT competency | ITC1 | 0.853 |
| | ITC3 | 0.720 |
| | ITC2 | 0.830 |
| | ITC4 | 0.729 |
| Relative advantage | RAD1 | 0.819 |
| | RAD2 | 0.879 |
| | RAD3 | 0.879 |
| | RAD4 | 0.733 |
| Top management support | TMS1 | 0.794 |
| | TMS2 | 0.872 |
| | TMS3 | 0.822 |
| | TMS4 | 0.790 |
| Acceptance | ACC1 | 0.705 |
| | ACC2 | 0.866 |
| | ACC3 | 0.869 |
| | ACC4 | 0.805 |
| Routinization | ROU1 | 0.779 |
| | ROU2 | 0.889 |
| | ROU3 | 0.846 |
| | ROU4 | 0.780 |
| Infusion | INF1 | 0.741 |
| | INF2 | 0.889 |
| | INF3 | 0.834 |
| | INF4 | 0.840 |

Note: DAD3 and OGC4 were deleted because of low loadings (< 0.7).

**Table 7. Results for convergent validity assessment.**

| Constructs | AVE > 0.50 |
|---|---|
| Compatibility | 0.688 |
| Complexity | 0.647 |
| Data demand | 0.738 |
| Incentives | 0.575 |
| Organizational culture | 0.747 |
| IT competency | 0.616 |
| Relative advantage | 0.688 |
| Top management support | 0.673 |
| Acceptance | 0.662 |
| Routinization | 0.680 |
| Infusion | 0.685 |

loading criterion, ii) Fornell and Larcker's (1981) criterion, and iii) Heterotrait-Monotrait ratio of correlations (HTMT).

The loadings for indicators for allocated latent variables should be greater than the loadings for all other latent variables under the cross-loading criterion. Chin [141] asserted that there must be no less than 0.1 variations in loading between latent variables. If each loading indicator is higher than the other constructs for its assigned construct, it can be claimed that the indicators are not replaceable with other constructs [131]. The results for the cross-loading criterion are presented in S1A Table in S1 Table. All the diagonal values of cross-loading for each block of the indicator are higher than other indicators. Hence, the data comply with the first discriminant validity assessment.

The second discriminant validity assessment is called Fornell–Larcker's criterion [142]. In this assessment, a latent variable explains more of the variance in its own indicators than other latent variables. The Fornell–Larcker's criterion can be analyzed by looking at the diagonal value of each latent variable. The results are depicted in S1B Table in S1 Table. The result indicates that the square root of AVE for each construct is greater than the inter-construct correlation. Thence, the result adds to the discriminant validity evidence of this study.

To further evaluate the discriminant validity using the HTMT criterion, a bootstrapping technique was performed. It is a resampling technique of original data in which the data are selected randomly to perform a calculation [143]. The result produces a slightly different value as the procedure is repeated to create a substantial number of samples. The typical bootstrapping iteration creates about 5,000 subsamples to estimate the standard error. The purpose of conducting the HTMT bootstrapping technique is to check whether the confidence interval's lower and upper bound contains the value of 1. If the confidence interval's range is found to stand between 1, it indicates that the data lack discriminant validity [131]. Likewise, if the confidence interval's range stands beyond 1, the two constructs are regarded as empirically distinct [132]. The results for the HTMT criterion are presented in S1C Table in S1 Table. The results of the HTMT bootstrapping indicate both the lower and upper bound of the confidence interval's range include a value of 1. As such, the discriminant validity of the data is established in accordance with the liberal criterion of HTMT inference.

## 6.2 Structural model assessment

Before assessing the structural model, it is critical to confirm that there are no issues with lateral collinearity [131]. This is because lateral collinearity (predictor-criterion collinearity)

occasionally deceives the results covertly, even when discriminant validity (vertical collinearity) is fulfilled [144]. Additionally, lateral collinearity problems may obscure the model's clear causal influence. This condition normally occurs when two variables that are believed to be causally connected evaluate the same construct. The collinearity (vertical) and lateral collinearity assessments are explained next.

In PLS-SEM analysis, a variance inflation factor (VIF) is used to demonstrate the existence of multicollinearity issues. A tolerance value of 0.20 or less and a VIF value of 5 or greater suggest the possibility of collinearity [145,146]. More precisely, the VIF level of an indicator of 5 suggests that the remaining formative indicators associated with the same construct account for 80% of its variance [132]. If the degree of collinearity is extremely strong, as shown by a VIF value of 5 or greater, it might be necessary to exclude one of the corresponding indicators [132]. However, the rest of the metrics must also correctly capture the content of the construct from a theoretical standpoint. The first VIF assessment revealed that no remaining indicator had violated the tolerance value, as Hair Jr, et al. [132] suggested, which is any score more than value 5.

The inner VIF values (compatibility, complexity, culture, data demand, IT competency, incentive, relative advantage, top management support) for all of the other independent variables that need to be tested for lateral multicollinearity were found to be less than 5, suggesting that lateral multicollinearity is not a problem in the analysis [132]. Table 8 summarizes the results for lateral collinearity assessment.

**6.2.1 Path coefficient.** The path coefficient for the relationship among latent variables is obtained through the SmartPLS [125] software bootstrapping technique. Inferential statistics (*t*-values) are obtained with bootstrap standard error. The *t*-values used for the significance was 1.645 for a one-tailed test and 1.96 for two-tailed tests [132]. In this study, one-tailed test was used, as suggested by Ramayah, et al. [131], because the hypotheses are directional.

In this study, 10 direct hypotheses, defined as H1 to H10, were developed. Results presented in Table 9 indicate that all relationships are significant with *t*-values above 1.645 at the 5% level ($\alpha = 0.05$; one-tailed test) except for relationships between Compatibility → Acceptance, IT Competency → Acceptance, Incentives → Acceptance, and Data demand → Acceptance, which were found to be nonsignificant. The results also showed that complexity ($\beta = -0.018$, $p = .007$), relative advantage ($\beta = 0.184$, $p < .005$), organizational culture ($\beta = 0.144$, $p = 0.021$), and top management support ($\beta = 0.28$, $p < .001$) are positively related to acceptance, which explains 45.2% of variances. Thus, H2, H3, H4, and H5, are supported.

Fig 4 illustrates the bootstrapping function's results with 5,000 iterations and using a one-tailed *t*-test with a significance level of 0.05. The $R^2$ value for acceptance is 0.452, indicating a substantial model, as suggested by Cohen (1988). The effect of acceptance on routinization indicates that the former ($\beta = 0.656$, $p < .001$) is positively related to the latter, explaining 43%

**Table 8. Results for lateral collinearity assessment.**

| Independent Variables | VIF $\leq$ 5.0 |
|---|---|
| Compatibility | 2.430 |
| Complexity | 1.207 |
| Organizational culture | 1.881 |
| Data demand | 1.542 |
| IT competency | 1.282 |
| Incentives | 1.247 |
| Relative advantage | 1.983 |
| Top management support | 1.786 |

**Table 9. Results for the path coefficient analysis.**

| Hypothesis | Relationship | Std Beta | Std Error | t-value | P Values | Decision |
|---|---|---|---|---|---|---|
| H1 | Compatibility -> Acceptance | −0.007 | 0.074 | 0.093 | .463 | Not supported |
| H2 | Complexity -> Acceptance | −0.018 | 0.074 | 2.441 | .008* | Supported |
| H3 | Relative advantage -> Acceptance | 0.184 | 0.072 | 2.552 | .004** | Supported |
| H4 | Organizational Culture -> Acceptance | 0.144 | 0.071 | 2.026 | .021* | Supported |
| H5 | Top management support -> Acceptance | 0.280 | 0.062 | 4.449 | 0.000*** | Supported |
| H6 | IT Competency -> Acceptance | 0.060 | 0.059 | 1.02 | .160 | Not supported |
| H7 | Incentives -> Acceptance | 0.040 | 0.048 | 0.048 | .205 | Not supported |
| H8 | Data Demand -> Acceptance | 0.107 | 0.072 | 1.488 | .065 | Not Supported |
| H9 | Acceptance -> Routinization | 0.656 | 0.037 | 17.486 | .000*** | Supported |
| H10 | Routinization -> Infusion | 0.711 | 0.030 | 23.502 | .000*** | Supported |

Note:

*$p < .05$,

**$p < .01$,

***$p < .001$;

t-values bootstrapped 5,000 times.

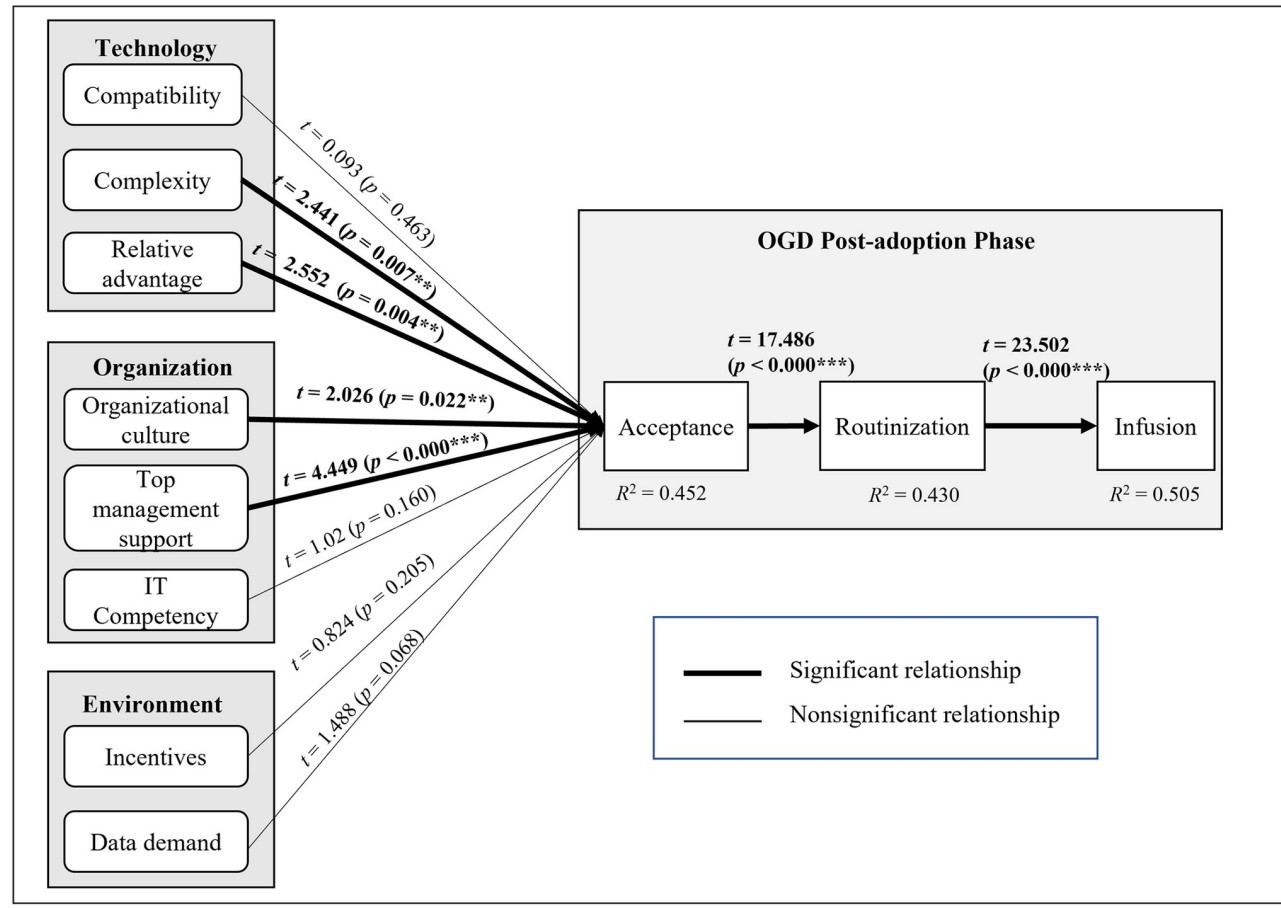

**Fig 4. Results for structural model assessment.**

of the variance of the latter. Simultaneously, the effect of routinization on infusion indicates that the former ($\beta = 0.711$, p < .001) is also positively related to the latter, explaining 50.5% of the variance of the latter. These results for endogenous constructs support this study's hypotheses H9 and H10.

## 7. Discussion

This research integrated three theoretical perspectives based on innovation adoption in organization literature: the innovation adoption process, post-adoption of innovation, and the TOE framework to build the OGD post-adoption framework in the public sector.

The first factor tested significantly to the OGD post-adoption is complexity (Hypothesis 1). Complexity was hypothesized to have a negative influence on OGD post-adoption. This notion implies that the data providers acknowledged the complexity of OGD publication. The finding is consistent with the literature, which states that the higher complexity of technology innovation will decrease the chance of its acceptance [129,147], thereby undermining OGD post-adoption. However, previous research by Yang and Wu [12] found that perceived effort, which has similar characteristics to complexity, was nonsignificant to government agencies' intention of OGD adoption. Yang and Wu [12] noted that the outcome was a result of adequate assistance from higher authorities to government entities in implementing OGD. Additionally, data providers frequently chose "low-hanging fruit" data in order to minimize publication complexity. Regardless, with the advent of high-intensity data technologies such as the blockchain, data providers may face complexity with OGD post-adoption in the near future. By that time, data providers need to be equipped with the requisite skills and knowledge to operate OGD with more challenging data structures. Hence, the stakeholders should pay attention to building the technical capacity to reduce the OGD publication complexity.

The second factor tested significantly to OGD post-adoption is the relative advantage (Hypothesis 3). Relative advantage is the extent to which an innovation is seen as superior to an idea and has the potential to benefit an organization [54]. In this study, the data providers argued that OGDs provide advantages and add value to organizations. OGD implementation has become one of the important components in most digital government strategic plans. Providing access to OGD means that as data providers, government agencies are strengthening the digital government strategic plans.

Organizational culture is the third-factor supporting OGD post-adoption (Hypothesis 4). In this study, the organizational culture refers to the government agencies' openness culture, which revolves around how data providers respond to managing and publicly sharing data [12]. Data providers agree that embedding data openness as an organizational culture will enhance OGD post-adoption. These days, the importance of OGD publication as organizational culture is more discernible as the government is striving to empower OGD as part of the digital government aspiration. Numerous government services and daily tasks are turning into the digital space, thereby allowing information and data to be managed with computer applications or ISs. Indirectly, the availability of non-confidential data in a machine-readable format sparks the data providers to make OGD publication an organizational culture.

The results suggest that the strongest antecedent for OGD post-adoption is the top management support represented by Hypothesis 5. Top management support has been recognized in many studies as a highly influential factor in innovation adoption in an organization [148]. Given that the OGD initiative was a mandated decision from a higher executive level in the federal government hierarchy, top management support, in this case, is apparent. This result is consistent with previous studies on OGD adoption in which top management support was observed as the most prominent factor [50].

As pointed out in Hypothesis 1, the relationship between compatibility and acceptance was shown to be insignificant. The findings suggest that the data's compatibility, operating practices, IT infrastructure, and agencies' values and beliefs do not influence OGD implementation in the post-adoption phase among government agencies. Centered on the source of authority given, the adoption of OGD in the public sector has been regarded as an obligation rather than a voluntary basis. The nationwide deployment has made the OGD a well-known initiative at various levels of government services. A lot of national ICT strategic planning has delineated OGD as one of the long-term initiatives to be focused on. Based on these reasons, it can be understood why compatibility is not a concern among government agencies because responding to the national agenda is more necessary. Within the same tones, Amade, et al. [149] and Junior, et al. [129] asserted that the compatibility of innovation is not statistically significant in the post-adoption phase. Amade, et al. [149] confirmed whether compatibility influences Mozambican institutions in an effort to continue using geographical information technologies. In contrast, Junior, et al. [129] emphasized that compatibility has no positive influence on enterprise resource planning diffusion among farmers in Brazil.

The second antecedent that is not supported by OGD post-adoption is IT competency (Hypothesis 6). It refers to the organization's technical capabilities, which include infrastructure and human resource capabilities in IT [150]. The findings suggest that government agencies have insufficient knowledge and technological resources to manage OGD post-adoption. Zhao and Fan [10] observed similar findings in which technical competency is a mandatory basis for managing OGD capacity. However, the data providers' IT competency level in operating OGD is still low and requires improvement. It is foreseen that more advanced skills are needed if the government agency intends to increase the OGD potential. Several scholars have suggested that public servants, regardless of their position and job title, should not just possess IT-related skills but instead build the ability to cope with emerging technologies [3,151]. In addition, Luna-Reyes and Najafabadi [25] posited that OGD management involving the combination of technology and data analysis expertise is still sparse. In order to strengthen the OGD implementation in public sector entities, highly skilled technical personnel and training are necessary [10].

Environmental aspects have outlined two measurements of factors that can influence the OGD post-adoption, namely the data demand and incentives. The relationship between data demand and OGD acceptance was tested through Hypothesis 7. The former refers to the request for certain datasets from the data provider's internal or external environs. It is a newly introduced factor in the OGD post-adoption framework to seek government agencies' feedback in responding to the external environment. The results indicate data providers' standpoint on data demand in which serving the needs of data consumers does not influence OGD post-adoption. Without data demand, OGD is merely supply-driven with less potential to be exploited into useful data products. A strong argument was highlighted in Ubaldi [3] when the researcher observed that the public sector spends more time on strategies rather than knowing the value creation of OGD by fulfilling the demand first. It is feared that disregarding the data demand will stifle the development of future inventions.

The final antecedent that is not supported OGD infusion is the incentives factor (Hypothesis 8). The findings indicate that incentive is not positively influenced by OGD infusion among government agencies. The incentives have the potential to be a positive factor in the future, but they need to be re-modeled. This is because the lack of motivation from the environment makes the OGD implementation highly supply-driven. In contrast, Ubaldi (2013) stated that providing incentives will foster full commitment among government agencies to sustain OGD implementation. However, from a different perspective, it is a great sign that the

data providers are committed to the OGD post-adoption and regard OGD publishing as part of their scope of operation without expecting anything in return.

This study's main research questions sought to answer: What are the factors that influence OGD post-adoption among the data providers? This research question was resolved by applying a quantitative method with the PLS-SEM as the factor analysis technique. Five factors from all three contexts (technology, organizational, environment) significantly influenced the post-adoption of OGD in the public sector. The factors are compatibility, relative advantage, top management support, culture, and data demand. The second research question this study tried to answer is how the data provider's readiness is to sustain the OGD implementation in their existing work structure. Based on the relatively significant associations among the independent variables and the post-adoption phase, it can be inferred that data providers are ready to accept, routinize, and infuse OGD into their existing work system. The data providers' readiness indicates that the OGD innovation will have a solid trajectory to be sustained and expanded in the future.

## 8. Managerial implication

This study's contributions are presented from three different perspectives, namely theoretical, conceptual, and practical. From a theoretical standpoint, this study contributes to the IT innovation adoption theory, particularly in the post-adoption phase, by experimenting with OGD initiatives. While most IT innovation adoption studies focused on adoption, this study promoted the post-adoption phase, namely the acceptance, routinization, and infusion stage. The post-adoption phase is important because the inability to secure profound use of IT innovation beyond the adoption phase could cause its abandonment [81]. To the best of the authors' knowledge, this study is the first to explore the post-adoption phase of OGD, particularly the routinization and infusion stage. Hence, this study presents empirical and practical evidence of OGD acceptance, routinization, and infusion in the post-adoption phase through ongoing experience and efforts in the Malaysian public sector. An important implication that this study tries to portray is that the lack of a post-adoption framework may hinder the government from planning the subsequent OGD implementation actions.

This study introduces new factors into the research model from the conceptual perspective, namely the incentives and data demand for the environmental context. However, the incentives and data demand factors show a nonsignificant contribution to the OGD post-adoption. Perhaps the environmental context can be tested in diverse socioeconomic or geopolitical statuses to see contrasting repercussions. Finally, from the practicality perspective, this study's outcome is anticipated to guide the policymakers in the government administration to use the assessments to sustain OGD implementation in the post-adoption phase from the data provider's perspective. The significant factors identified in this study can act as a catalyst to foster more seamless OGD acceptance, routinization, and infusion.

## 9. Limitations and future works

Like any other research, this study suffers from a few limitations, which indirectly offer some room for future research opportunities. First, the eight post-adoption factors within the TOE context covered all factors for innovation adoption in an organization. However, there is another context that is not covered in TOE, such as the human context. Therefore, the opportunity to explore other contributing factors for the post-adoption of OGD in an organization is in need. Additionally, replicating the study in different socioeconomic or geopolitical statuses would further enrich the findings. Secondly, a cross-sectional study was employed in this study in which the data were collected in a specific timeframe. Hence, employing a

longitudinal study would extend the direction of OGD in the post-adoption phase. Third, this study centered on the Malaysian public sector and its surrounding nature. The existence of the private sector or non-government organizations as OGD providers is yet to be explored. The individual factor is ignored intentionally because this study's objective was to explore OGD at an organizational level of analysis. Finally, all the factors were measured using a self-administered instrument, which may be exposed to self-reported bias. Hence, using multiple measurement methods may avoid such potential occurrences in the future. In light of such a study, subsequent research must draw on the essential outcomes of this research.

For future studies, the developed OGD post-adoption framework can be extended to other IS/IT innovation research to enrich post-adoption research. OGD users and OGD beneficiaries have a vital role in making OGD sustainable in the OGD ecosystem. It is conceivable that research from the view of these roles can be performed to understand how OGD is being used and impacted their life. In this way, government agencies can improve the OGD publication in terms of data quality.

## 10. Conclusions

This study intended to investigate the OGD post-adoption among data providers in the public sector. Guided by the TOE framework and innovation adoption theory, the OGD post-adoption framework in the public sector was realized. The innovation adoption theory is the basic foundation for understanding the phases involved in innovation diffusion in an organization. It can be inferred that the post-adoption phase is the highest phase for innovation adoption in an organization. As posited by many scholars, failure to reach the post-adoption phase will result in the adoption of innovations reaching an impasse. A comprehensive theory or framework that can be used to describe the entire process of open data is limited; this includes the goal of OGD, the innovation process, and the results or impacts of OGD. In that sense, research on OGD post-adoption is essential to enrich the OGD body of knowledge. Ergo, it is imperative to assess an innovation's acceptance, routinization, and infusion stage to chart a better path forward. Deliberately, this study also highlights the need for more post-adoption innovation research in determining the next steps subsequent to adoption. Following the aforementioned cues, OGD innovation must take alternative and practical measures to ensure its implementation within the government. The post-adoption phase appears to justify the current OGD implementation in the Malaysian public sector and indirectly signifies OGD implementation maturity among data providers. Three general contexts (technology, organization, environment) were investigated to determine whether the factors significantly influence OGD post-adoption among data providers. The investigation was conducted through a survey instrument disseminated among the data providers in the Malaysian public sector. The data collected were analyzed using established research methods in a quantitative study. The results exhibit that top management support, organizational culture, relative advantage, and complexity are the factors that influence OGD in the post-adoption phase. Consequently, the data providers are expected to support sustaining OGD with the factors emphasized by the stakeholders. In this way, the OGD implementation way forward can be determined and remain pertinent in the years to come.

## Supporting information

**S1 Table. The results for measurement model assessment.**
(DOCX)

**S1 Appendix. Survey questionnaire for open government data post-adoption factors among data providers.**
(DOCX)

**S2 Appendix. Interview scripts for Semi-structured interview during preliminary study.**
(DOCX)

## Acknowledgments

The author would like to acknowledge the support from the Digital Government Division of the Malaysian Administrative Modernization and Management Planning Unit (MAMPU) through various insights and feedback given towards completing this study.

## Author Contributions

**Conceptualization:** Mimi Nurakmal Mustapa.

**Investigation:** Mimi Nurakmal Mustapa.

**Supervision:** Suraya Hamid, Fariza Hanum Md Nasaruddin.

**Validation:** Suraya Hamid, Fariza Hanum Md Nasaruddin.

**Writing – original draft:** Mimi Nurakmal Mustapa.

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
