## [Decision Letter · Decision Letter 0]

10 Jun 2022

PONE-D-22-00326Factors influencing open government data post-adoption in the public sector: The perspective of data providersPLOS ONE

Dear Dr. Hamid,

Thank you for submitting your manuscript to PLOS ONE. After careful consideration, we feel that it has merit but does not fully meet PLOS ONE’s publication criteria as it currently stands. Therefore, we invite you to submit a revised version of the manuscript that addresses the points raised during the review process.

We look forward to receiving your revised manuscript.

Kind regards,

Dragan Pamucar

Academic Editor

PLOS ONE

Journal Requirements:

2. "Please provide additional details regarding participant consent to collect personal data, including email addresses, names, or phone numbers. In the Methods section, please ensure that you have specified how consent was obtained and how the study met relevant personal data and privacy laws. If data were collected anonymously, please include this information.”

3. We note that you have stated that you will provide repository information for your data at acceptance. Should your manuscript be accepted for publication, we will hold it until you provide the relevant accession numbers or DOIs necessary to access your data. If you wish to make changes to your Data Availability statement, please describe these changes in your cover letter and we will update your Data Availability statement to reflect the information you provide

Reviewers' comments:

Reviewer's Responses to Questions

**Comments to the Author**

1. Is the manuscript technically sound, and do the data support the conclusions?

Reviewer #1: Partly

Reviewer #2: Yes

2. Has the statistical analysis been performed appropriately and rigorously? 

Reviewer #1: Yes

Reviewer #2: Yes

3. Have the authors made all data underlying the findings in their manuscript fully available?

Reviewer #1: Yes

Reviewer #2: Yes

4. Is the manuscript presented in an intelligible fashion and written in standard English?

Reviewer #1: Yes

Reviewer #2: Yes

5. Review Comments to the Author

Reviewer #1: Given my background in information systems, the topic of the paper clearly and closely matches my research area. This paper is well-written and addresses the gap. The results show some novel insights including factors on OGD post-adoption phase. However, the article in its current form is not ready for publication unless the improvements are made:

1. There are no citations in 49-56. Please see.

2. In line 58, benefits and line 59 objectives are described. What are the benefits? Benefits or objectives. Please be consistent.

3. Do you find all the constructs reflective? How and why? Why is there no need to use any formative construct? Is there any statistical test that can give you confidence that the constructs are reflective? Though there are specific procedural and logical criteria to measure or consider a construct reflective or formative, a test can be used to statistically test a construct reflective or formative. Perform that test and show the results.

4. Citation format in line 75 and 76 are incorrect.

5. In reference in line 77 also?

6. Exploitation stage refers to….?

7. Introduction Section is lacking significance and expected contribution of the study.

8. The statement “The OGD is a combination of innovation, methodology, and organizational-level initiative that operates ideally in a data-sharing ecosystem” needs an authentic reference.

9. How do you differentiate between implementation and post-adoption? Why is there a need to explicitly write “post-adoption”? What are the advantages you get while writing post-adoption over implementation?

10. Better to write, “Most of the time, these data are not accessible to the public” instead of Most of the time, these data are stored in a way that is inaccessible to the public...

11. Reference is incorrect “58. Tornatzky LG, Fleischer M. Processes of Technological Innovation. Lexington, MA.: Lexington Books; 1990.”. Please correct it.

12. Line 208, how do you determine that it is the highest? Groups as a unit of analysis is lowest? Countries as a unit of analysis is not highest? Please explain highest.

13. Line 215, How can you determine that TOE is the most recognized attempt…? Any reference? Strong justification is necessary here.

14. Section 3, a real need to join two theories is not clear.

15. Reference 45. “Lewin K. Frontiers in group dynamics: Concept, method and reality in social science; social equilibria and social change. Human relations. 1947;1(1):5-41.” is incorrect. The paper the study number 44 is referring is different from your study. Insert correct reference.

16. Lines 255-258, selection of constructs to put in the framework is not clear. Which construct from which theory/model?

17. Line 173-175. First, as mentioned above the reference [45] is incorrect. Please see “IT innovation adoption in the government sector: identifying the critical success factors” study. Second, these studies [44, 46-52] might have categorized the innovation adoption process into phases but no study had claimed that our/this study was centered around Lewin’s [45] study/concept of change. Need to revise the sentence or provide correct references/studies that are centered on lewin’s [45] study.

18. Section 4.1 and 4.2. The base and evidences are insufficient to suggest hypothesis. There are several other studies. Strengthen the sections.

19. Section 4.3. No base study/references of studies in OGD domain in suggesting hypothesis. Make this section more strong.

20. No reference in Section 4.7 except only 1. There are so many other studies which can be referred in suggesting the hypothesis.

21. In Section 4.8, there are several references in OGD that can be inserted between lines 388-398. Please insert relevant references.

22. In Section 4.9 and 4.10, grounds are too short to propose hypothesis. Further, no relevant references from OGD literature in respective sections. Remove all those references outside the domain of OGD as OGD literature itself is also containing a lot of grounds in proposing hypothesis. Moreover, paragraphs of relevant sections are too short. There must be some consistency of length of paragraphs in proposing hypothesis as some paragraph are too short or too long.

23. The word ‘several’ with ‘only’ is not appropriate here in Line 494.

24. Grammatical mistakes in lines 506-509.

25. In Line 518, Please add those paid tools which were used to collect data.

26. Sampling technique and the reasons to choose sampling technique were not elaborated. Please elaborate.

27. Table 3, transformation or adaptation of items from previous literature is seriously questionable. Suppose, the items of Routinization. Study 89 is a conference paper. Moreover, items of Routinization used in current study are not truly reflecting the items taken from Studies 105 and 107. Similarly, it is happening in items of Infusion. Moreover, item DAD1 is grammatically incorrect/incomplete. The validity of the questionnaire is questionable? Please provide mapping of all items of all constructs (Table 3) with the previous literature within or outside the OGD domain.

28. Which constructs are formative and which are reflective? Please mention/elaborate/justify.

29. No issue is found in results of Measurement and Structural model. However, a valid and reliable questionnaire avoids rolling of errors in result, discussion, and conclusion which is not true in the current study.

30. Line 757-758, “this result is consistent with….”. In fact, the result is not consistent with reference 32 because the study is related to the adoption of OGD at the firm level from the data use perspective instead of data providers’ perspective (as your main research question is “What are the factors that influence OGD post-adoption among the data providers?)”.

31. Overall, the manuscript has a large references outside the domain of OGD, even a large number of studies conducted in OGD.

Reviewer #2: Adoption VS Implementation are two different terms. But are used interchangeably in the manuscript. Please synchronizes and define each in manuscript.

Please define DOI,TAM and TPB before using it.

How the factors are selected for theoretical framework. Please justify in manuscript.

Table 3 have some factor items with our references. Are these items developed by the author. If so, instrument validation is performed????

Is Instrument validation and pilot study is performed before actual data collection. if so, who are experts for instruments validation.

How the interview data is processed. Please justify in manuscript.

Please provide some interview scripts in manuscript for the reader.

Most of the reference are older than 2019.Please include latest literature published during last 2 years.

Hoe many companies were contacted for data collection??? what was the response rate? what are the number of respondents for actual data analysis after data screening etc.????

Figure 1 to 3 not readable. Please adjust the figures for more readability.

Some Grammatical mistake need to be corrected.

6. PLOS authors have the option to publish the peer review history of their article (what does this mean?). If published, this will include your full peer review and any attached files.

Reviewer #1: No

Reviewer #2: **Yes: **Hafiz Muhammad Faisal Shehzad

---

## [Author Response · Author response to Decision Letter 0]

31 Aug 2022

Reviewer 1

Given my background in information systems, the topic of the paper clearly and closely matches my research area. This paper is well-written and addresses the gap. The results show some novel insights including factors on OGD post-adoption phase. However, the article in its current form is not ready for publication unless the improvements are made:

1. There are no citations in 49-56. Please see.

RESPONSE: Thank you for your suggestion. We have rewritten the sentences and citation has been added (line 51). 

2. In line 58, benefits and line 59 objectives are described. What are the benefits? Benefits or objectives. Please be consistent.

RESPONSE: Thank you for your suggestion. We have changed the term to benefits (line 55).

3. Do you find all the constructs reflective? How and why? Why is there no need to use any formative construct? Is there any statistical test that can give you confidence that the constructs are reflective? Though there are specific procedural and logical criteria to measure or consider a construct reflective or formative, a test can be used to statistically test a construct reflective or formative. Perform that test and show the results.

RESPONSE: Thank you for providing these insights. Constructs were determined as reflective or formative using Confirmatory Tetrad Analysis (CTA). The CTA has been performed and added in the analysis section (starting at line 759)

4. Citation format in line 75 and 76 are incorrect.

RESPONSE: We agree with you and have incorporated this suggestion throughout our paper. The in-citation format at line 73 (previously 75 and 76) have been corrected.

5. In reference in line 77 also?

RESPONSE: The in-citation format at line 74 (previously 76) have been corrected.

6. Exploitation stage refers to….?.

RESPONSE: We would like to admit our mistake on the wrong choice of word. The ‘exploitation stage’ term has been changed to ‘initiation stage’ (line 83).

7. Introduction Section is lacking significance and expected contribution of the study. 

RESPONSE: Thank you for your insights. We have redrafted the introduction section. We think these changes are now better than the previous draft. We hope that you agree.

8. The statement “The OGD is a combination of innovation, methodology, and organizational-level initiative that operates ideally in a data-sharing ecosystem” needs an authentic reference.

RESPONSE: We agree with your assessment; hence we have removed the sentences and rewritten the paragraph. Changes can be seen at lines 131-137 

9. How do you differentiate between implementation and post-adoption? Why is there a need to explicitly write “post-adoption”? What are the advantages you get while writing post-adoption over implementation?

RESPONSE: 

How do you differentiate between implementation and post-adoption?

i. We have added the explanation between implementation and post-adoption at this statements (line 226-229) in the manuscript:

Implementation is one of the post-adoption activities, while post-adoption is the phase that happens after innovation has been decided to adopt during the adoption (decision) phase (Rogers, 1995; Damanpour, 2006). 

Why is there a need to explicitly write “post-adoption”?

i. We have added these explanation to explicitly use the term ‘post-adoption’ at line 232-239.

Following the innovation adoption process, this study resorted to using the post-adoption term to identify the consequences of OGD in data providers’ environments. The question raised by Rogers (1983), “Why Haven’t Consequences Been Studied More?”, suggested that post-adoption is a pro-found area of study. The fact that there are various desirable or undesirable effects on adopters or the social system, thus widens the post-adoption study perspective. The undesirable consequences that could happen in the post-adoption phase include decommissioning, stagnant, discontinuation of innovation, and the like that goes beyond pro-innovation bias (Rogers, 1983). 

What are the advantages you get while writing post-adoption over implementation?

i. Some advantage of using the post-adoption term is that the:

ii. ‘Successful and beneficial adoption of an innovation is acknowledged when the innovation is put into practice and integrated into the organization’ 

iii. The statement is added at line (252-254).

10. Better to write, “Most of the time, these data are not accessible to the public” instead of Most of the time, these data are stored in a way that is inaccessible to the public...

RESPONSE: Thank you for your suggestion. However, we have removed this sentence for it has no context to the previous sentences.

11. Reference is incorrect “58. Tornatzky LG, Fleischer M. Processes of Technological Innovation. Lexington, MA.: Lexington Books; 1990.”. Please correct it.

RESPONSE: Corrected the reference to “67. Tornatzky L, Fleischer M. Processes of Technological Innovation. Lexington, MA.: Lexington Books; 1990.”

12. Line 208, how do you determine that it is the highest? Groups as a unit of analysis is lowest? Countries as a unit of analysis is not highest? Please explain highest.

RESPONSE: Thank you for highlighting this. We admit our mistake in this statement. A country analysis is the highest unit of analysis. We have removed the statement from the manuscript.

13. Line 215, How can you determine that TOE is the most recognized attempt…? Any reference? Strong justification is necessary here.

RESPONSE: We proposed this statement because, according to some studies, the TOE framework’s proposed generic factors and offer more insightful lenses for examining users’ perceptions of particular technologies, making it valid, most dominant, and specific for enterprise-context adoption (Gangwar et al., 2014, Al-Natour and Benbasat, 2009). Furthermore, the T-O-E framework addresses the requirement for more socioeconomic strides and has received more substantial theoretical and empirical backing in the IS area than many other adoption models (Awa, 2017). According to Zhu et al. 2003 theoretical evaluation, the TOE framework is more important than Rogers’s (1983) Innovation Diffusion Theory.

The justification statements have been added in the manuscript at line (271-282).

14. Section 3, a real need to join two theories is not clear.

RESPONSE: We have added the explanation of joining the two theories at line 305-312.

15. Reference 45. “Lewin K. Frontiers in group dynamics: Concept, method and reality in social science; social equilibria and social change. Human relations. 1947;1(1):5-41.” is incorrect. The paper the study number 44 is referring is different from your study. Insert correct reference.

RESPONSE: Thank you for highlighting this mistake. We have corrected the reference to “57. Lewin K. Group Decision and Social Change. Readings in Social Psychology. 1952; 3(1):197-211.”

16. Lines 255-258, selection of constructs to put in the framework is not clear. Which construct from which theory/model?

RESPONSE: Thank you for this suggestion. We have added the organization of each construct and theory at line 325-333.

17. Line 173-175. First, as mentioned above the reference [45] is incorrect. Please see “IT innovation adoption in the government sector: identifying the critical success factors” study. Second, these studies [44, 46-52] might have categorized the innovation adoption process into phases but no study had claimed that our/this study was centered around Lewin’s [45] study/concept of change. Need to revise the sentence or provide correct references/studies that are centered on lewin’s [45] study.

RESPONSE: We agree with your assessment and we are glad that our paper has been reviewed by prominent scholars. We have rewritten the sentences, elaborated each study accordingly, and removed studies unrelated to Lewin’s (1947). (Lines 209-213)

18. Section 4.1 and 4.2. The base and evidences are insufficient to suggest hypothesis. There are several other studies. Strengthen the sections.

RESPONSE: Thank you for the suggestions. We have strengthened sections 4.1 and 4.2 with evidence from previous studies.

19. Section 4.3. No base study/references of studies in OGD domain in suggesting hypothesis. Make this section more strong.

RESPONSE: Thank you for this suggestion. Section 4.3 has been rewritten by adding more references to strengthen the study.

20. No reference in Section 4.7 except only 1. There are so many other studies which can be referred in suggesting the hypothesis.

RESPONSE: Thank you for this suggestion. We have rewritten Section 4.7 with more references of the context and to be more in line with your comments.

21. In Section 4.8, there are several references in OGD that can be inserted between lines 388-398. Please insert relevant references.

RESPONSE: Thank you for this suggestion. We have added relevant references in OGD at Lines 513 -521.

22. In Section 4.9 and 4.10, grounds are too short to propose hypothesis. Further, no relevant references from OGD literature in respective sections. Remove all those references outside the domain of OGD as OGD literature itself is also containing a lot of grounds in proposing hypothesis. Moreover, paragraphs of relevant sections are too short. There must be some consistency of length of paragraphs in proposing hypothesis as some paragraph are too short or too long.

RESPONSE: Thank you for your suggestions. We have redrafted sections 4.9 and 4.10 with relevant references from OGD literature and removed references outside of the OGD domain. However, we retain the references that we use as the theoretical lens of the study.

23. The word ‘several’ with ‘only’ is not appropriate here in Line 494.

RESPONSE: Thank you for your concern. We have rewrite the phrase to ‘During the early phase of OGD adoption, there were less than two hundred datasets published in the government open data portal’ (Line 667).

24. Grammatical mistakes in lines 506-509.

RESPONSE: Thank you for your suggestion. We have corrected the grammar mistakes at the mentioned line. The new lines for the statement are at 685-689.

25. In Line 518, Please add those paid tools which were used to collect data.

RESPONSE: We have added the name of the paid tools used in the study at line 700.

26. Sampling technique and the reasons to choose sampling technique were not elaborated. Please elaborate.

RESPONSE: Thank you for highlighting this. We have elaborated the sampling technique at lines 674-682.

27. Table 3, transformation or adaptation of items from previous literature is seriously questionable. Suppose, the items of Routinization. Study 89 is a conference paper. Moreover, items of Routinization used in current study are not truly reflecting the items taken from Studies 105 and 107. Similarly, it is happening in items of Infusion. Moreover, item DAD1 is grammatically incorrect/incomplete. The validity of the questionnaire is questionable? Please provide mapping of all items of all constructs (Table 3) with the previous literature within or outside the OGD domain.

RESPONSE: 

i. Thank you for providing these insights. At the moment, there is no study about routinization and infusion of OGD. To the best of the authors’ knowledge, this study is the first to explore the post-adoption phase of OGD, particularly the routinization and infusion stage (we add this explanation at line 573-574 and 604-605). 

ii. Hence, the references are mostly outside of the OGD domain. The question items for routinization and infusion were self-developed by authors through the concept of routinization and infusion from studies in other fields. However, we have taken the reviewer's advice and remove the Studies 105 and 107, instead replace with a ‘self-developed’ indication. Other items’ citations have been updated within or outside the OGD domain.

iii. To strengthen the validity of the survey, the survey instrument underwent content validity procedure and pilot test before the main study was conducted (we add this explanation at line 697-699). However, we did not cover the content validity procedure in this paper due to its lengthy steps and not included as one of the scope of this paper.

iv. The survey instrument was built in two language versions (Malay and English). Most of the respondents choose to answer in the Malay version, thus the grammar mistake was overlooked. Thank you for pointing out the grammar error for item DAD1. We will be more careful in selecting our proofreading services.

28. Which constructs are formative and which are reflective? Please mention/elaborate/justify.

RESPONSE: Thank you for highlighting this. The formative and reflective constructs have been elaborated through the Confirmatory Tetrad Analysis (CTA) starting at line 2242 until 2363.

29. No issue is found in results of Measurement and Structural model. However, a valid and reliable questionnaire avoids rolling of errors in result, discussion, and conclusion which is not true in the current study.

RESPONSE: Thank you for providing these insights. Please see point #27 above. 

30. Line 757-758, “this result is consistent with….”. In fact, the result is not consistent with reference 32 because the study is related to the adoption of OGD at the firm level from the data use perspective instead of data providers’ perspective (as your main research question is “What are the factors that influence OGD post-adoption among the data providers?)”.

RESPONSE: Thank you for providing these insights. We agree with your assessment, therefore, we have removed reference 32 (now reference no. 44).

31. Overall, the manuscript has a large references outside the domain of OGD, even a large number of studies conducted in OGD.

RESPONSE: Thank you for providing these insights. We have added the latest related references of OGD in the manuscript and removed some of the unrelated references to OGD.

 

Reviewer 2

1. Adoption VS Implementation are two different terms. But are used interchangeably in the manuscript. Please synchronizes and define each in manuscript.

RESPONSE: We have synchronized the term adoption and implementation throughout the manuscript. We have defined the implementation term at line 227 and adoption term in Table 2 (line 242).

2. Please define DOI,TAM and TPB before using it.

RESPONSE: Thank you for this suggestion. We have added the definition of DOI, TAM and TPB at lines 166 to 72.

3. How the factors are selected for theoretical framework. Please justify in manuscript.

RESPONSE: Thank you for stressing this matter. We have added the organization of each construct and theory at lines 325-332.

4. Table 3 have some factor items with our references. Are these items developed by the author. If so, instrument validation is performed????

RESPONSE: Yes, some of the items were developed by the authors such as the items for ‘Data demand’ and ‘Incentive’ constructs, because these were the new constructs discovered during the interview. An instrument validation was performed through a content validity procedure. However, due to its lengthy steps, the content validity procedure was not covered in this manuscript.

5. Is Instrument validation and pilot study is performed before actual data collection. if so, who are experts for instruments validation.

RESPONSE: The instrument validation was performed through a content validity procedure by eight(8) Information System experts/lecturers from various universities in Malaysia. The instrument was also tested in a pilot study before the primary study. 

6. How the interview data is processed. Please justify in manuscript.

RESPONSE: We have added the method to process the interviews’ transcriptions in lines 627- 629.

7. Please provide some interview scripts in manuscript for the reader.

RESPONSE: Thank you for this suggestion. However, we think that the interview scripts are more suitable to be put at the Appendix section because it is not the main concern of the paper. Hence we have included the interview scripts as S2_Appendix.

8. Most of the reference are older than 2019. Please include latest literature published during last 2 years.

RESPONSE: Thank you for this concern. We have added the latest significant references from the last 2 years.

9. How many companies were contacted for data collection??? what was the response rate? what are the number of respondents for actual data analysis after data screening etc.????

RESPONSE: There were 671 government agencies contacted for the study (mentioned in line 682). The response rate was 40% (mentioned in line 730). The number of respondents for data analysis were 266 after the data screening process (mentioned in line 728- 731)

10. Figure 1 to 3 not readable. Please adjust the figures for more readability.

RESPONSE: We would like to apologize for the inconvenience. The figures have been adjusted for more readability.

11. Some Grammatical mistake need to be corrected.

RESPONSE: Thank you for your suggestions. The grammar has been checked throughout the entire manuscript and we have corrected the mistake accordingly.

---

## [Decision Letter · Decision Letter 1]

19 Sep 2022

PONE-D-22-00326R1Factors influencing open government data post-adoption in the public sector: The perspective of data providersPLOS ONE

Dear Dr. Hamid,

Thank you for submitting your manuscript to PLOS ONE. After careful consideration, we feel that it has merit but does not fully meet PLOS ONE’s publication criteria as it currently stands. Therefore, we invite you to submit a revised version of the manuscript that addresses the points raised during the review process.

We look forward to receiving your revised manuscript.

Kind regards,

Dragan Pamucar

Academic Editor

PLOS ONE

Journal Requirements:

Reviewers' comments:

Reviewer's Responses to Questions

**Comments to the Author**

1. If the authors have adequately addressed your comments raised in a previous round of review and you feel that this manuscript is now acceptable for publication, you may indicate that here to bypass the “Comments to the Author” section, enter your conflict of interest statement in the “Confidential to Editor” section, and submit your "Accept" recommendation.

Reviewer #1: All comments have been addressed

Reviewer #2: All comments have been addressed

2. Is the manuscript technically sound, and do the data support the conclusions?

Reviewer #1: Yes

Reviewer #2: Yes

3. Has the statistical analysis been performed appropriately and rigorously? 

Reviewer #1: Yes

Reviewer #2: Yes

4. Have the authors made all data underlying the findings in their manuscript fully available?

Reviewer #1: Yes

Reviewer #2: Yes

5. Is the manuscript presented in an intelligible fashion and written in standard English?

Reviewer #1: Yes

Reviewer #2: Yes

6. Review Comments to the Author

Reviewer #1: 1. “Yet, the underlying factors influencing OGD after the adoption phase are scarce”. The factors are scarce OR studies are scarce? Please confirm the intended meaning of the sentence.

2. In response to our previous observation No. 13 i.e. “Line 215, How can you determine that TOE is the most recognized attempt…? Any reference? Strong justification is necessary here.”, You may add two references, particularly in the open data context, here (1) Modeling of Open Government Data for Public Sector Organizations Using the Potential Theories and Determinants—A Systematic Review (2) Factors Affecting Performance of Open Government Data Initiatives: A Multi-Method Approach Using Sem and FSQCA.

Reviewer #2: (No Response)

7. PLOS authors have the option to publish the peer review history of their article (what does this mean?). If published, this will include your full peer review and any attached files.

Reviewer #1: **Yes: **Muhammad Mahboob Khurshid

Reviewer #2: **Yes: **Hafiz Muhammad Faisal Shehzad

---

## [Author Response · Author response to Decision Letter 1]

25 Sep 2022

Reviewer #1: 

1. “Yet, the underlying factors influencing OGD after the adoption phase are scarce”. The factors are scarce OR studies are scarce? Please confirm the intended meaning of the sentence.

Response: Thank you for highlighting this. We have clarified the statement at line 30 as follows.

“Yet, studies on the underlying factors influencing OGD after the adoption phase are scarce”.

2. In response to our previous observation No. 13 i.e. “Line 215, How can you determine that TOE is the most recognized attempt…? Any reference? Strong justification is necessary here.”, You may add two references, particularly in the open data context, here (1) Modeling of Open Government Data for Public Sector Organizations Using the Potential Theories and Determinants—A Systematic Review (2) Factors Affecting Performance of Open Government Data Initiatives: A Multi-Method Approach Using Sem and FSQCA.

Response: Thank you for your suggestion. We have added the two references on lines 284-296 to justify the TOE is the most utilized theory in IS/IT adoption study. Apologize for overlooking these highly relevant papers to our study. 

Journal Requirement:

There are two new references added to the manuscript as follows:

5. Hossain MA, Rahman S, Quaddus M, Hooi E, Olanrewaju A-S. Factors Affecting Performance of Open Government Data Initiatives: A Multi-Method Approach Using Sem and FSQCA. Journal of Organizational Computing and Electronic Commerce. 2021;31(4):300-19.

72. Khurshid MM, Zakaria NH, Rashid A, Ahmad MN, Arfeen MI, Faisal Shehzad HM, editors. Modeling of open government data for public sector organizations using the potential theories and determinants—a systematic review. Informatics; 2020: MDPI.

---

## [Decision Letter · Decision Letter 2]

17 Oct 2022

Factors influencing open government data post-adoption in the public sector: The perspective of data providers

PONE-D-22-00326R2

Dear Dr. Hamid,

We’re pleased to inform you that your manuscript has been judged scientifically suitable for publication and will be formally accepted for publication once it meets all outstanding technical requirements.

Kind regards,

Dragan Pamucar

Academic Editor

PLOS ONE

Additional Editor Comments (optional):

Reviewers' comments:

Reviewer's Responses to Questions

**Comments to the Author**

1. If the authors have adequately addressed your comments raised in a previous round of review and you feel that this manuscript is now acceptable for publication, you may indicate that here to bypass the “Comments to the Author” section, enter your conflict of interest statement in the “Confidential to Editor” section, and submit your "Accept" recommendation.

Reviewer #1: All comments have been addressed

2. Is the manuscript technically sound, and do the data support the conclusions?

Reviewer #1: Yes

3. Has the statistical analysis been performed appropriately and rigorously? 

Reviewer #1: Yes

4. Have the authors made all data underlying the findings in their manuscript fully available?

Reviewer #1: Yes

5. Is the manuscript presented in an intelligible fashion and written in standard English?

Reviewer #1: Yes

6. Review Comments to the Author

Reviewer #1: (No Response)

7. PLOS authors have the option to publish the peer review history of their article (what does this mean?). If published, this will include your full peer review and any attached files.

Reviewer #1: **Yes: **Muhammad Mahboob Khurshid

---

## [Editor Report · Acceptance letter]

20 Oct 2022

PONE-D-22-00326R2 

Factors influencing open government data post-adoption in the public sector: The perspective of data providers 

Dear Dr. Hamid:

I'm pleased to inform you that your manuscript has been deemed suitable for publication in PLOS ONE. Congratulations! Your manuscript is now with our production department. 

Kind regards, 

on behalf of

Dr. Dragan Pamucar 

Academic Editor

PLOS ONE